# On student-teacher deviations in distillation: does it pay to disobey?

**Vaishnavh Nagarajan**
Google Research
vaishnavh@google.com

**Aditya Krishna Menon**
Google Research
adityakmenon@google.com

**Srinadh Bhojanapalli**
Google Research
bsrinadh@google.com

**Hossein Mobahi**
Google Research
hmobahi@google.com

**Sanjiv Kumar**
Google Research
sanjivk@google.com

## Abstract

Knowledge distillation (KD) has been widely used to improve the test accuracy of a "student" network, by training it to mimic the soft probabilities of a trained "teacher" network. Yet, it has been shown in recent work that, despite being trained to fit the teacher's probabilities, the student may not only significantly deviate from the teacher probabilities, but may also outdo than the teacher in performance. Our work aims to reconcile this seemingly paradoxical observation. Specifically, we characterize the precise nature of the student-teacher deviations, and argue how they *can* co-occur with better generalization. First, through experiments on image and language data, we identify that these probability deviations correspond to the student systematically *exaggerating* the confidence levels of the teacher. Next, we theoretically and empirically establish another form of exaggeration in some simple settings: KD exaggerates the implicit bias of gradient descent in converging faster along the top eigendirections of the data. Finally, we tie these two observations together: we demonstrate that the exaggerated bias of KD can simultaneously result in both (a) the exaggeration of confidence and (b) the improved generalization of the student, thus offering a resolution to the apparent paradox. Our analysis brings existing theory and practice closer by considering the role of gradient descent in KD and by demonstrating the exaggerated bias effect in both theoretical and empirical settings.

## 1 Introduction

In knowledge distillation (KD) [6, 17], one trains a small "student" model to match the predicted soft label distribution of a large "teacher" model, rather than the one-hot labels that the training data originally came with. This has emerged as a highly effective model compression technique, and has inspired an actively developing literature that has sought to explore applications of distillation to various settings [41, 13, 52], design more effective variants [44, 3, 38, 5], and better understand theoretically when and why distillation is effective [32, 40, 35, 2, 8, 34, 10, 43, 25, 14, 39].

On paper, distillation is intended to help by transferring the soft probabilities of the (one-hot loss trained) teacher over to the student. Intuitively, we would desire this transfer to be perfect: the more a student fails to match the teacher's probabilities, the more we expect its performance to suffer. After all, in the extreme case of a student that simply outputs uniformly random labels, the student's performance would be as poor as it can get.

37th Conference on Neural Information Processing Systems (NeurIPS 2023).

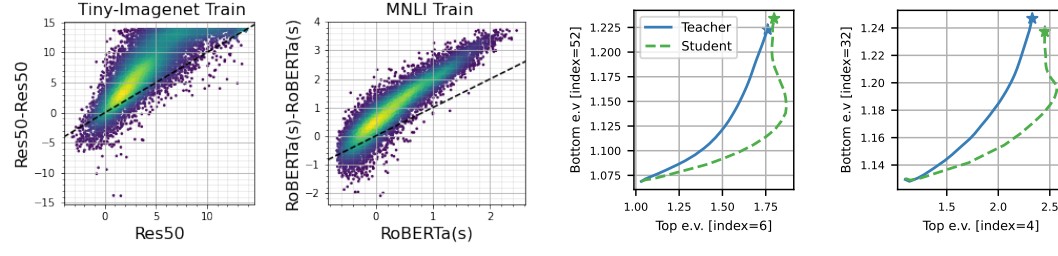

(a) Exaggeration of confidence      (b) Exaggeration of implicit bias

Figure 1: **(a): Distilled student exaggerates confidence of one-hot-loss trained teacher.** For each training sample $(x, y)$, we plot $X = \phi(p_{y^{\text{te}}}^{\text{te}}(x))$ versus $Y = \phi(p_{y^{\text{te}}}^{\text{st}}(x))$, which are the teacher and student probabilities on the teacher's predicted label $y^{\text{te}}$, transformed monotonically by $\phi(u) = \log[u/(1-u)]$. Note that this is a density plot where higher the brightness, higher the number of datapoints with that $X$ and $Y$ value. We find that the distilled student predictions deviate from the $X = Y$ line by either underfitting teacher's low confidence points (i.e., we find $Y \leq X$ for small $X$) and/or overfitting teacher's high confidence points (i.e., $Y \geq X$ for large $X$). See §3 for details. **(b) Distillation exaggerates implicit bias of one-hot gradient descent training.** We consider an MLP trained on an MNIST-based dataset. Each plot shows the time-evolution of the $\ell_2$ norm of the first layer parameters projected onto two randomly picked eigendirections; the $\star$'s corresponds to the final parameters. First observe that the one-hot-trained teacher moves faster towards its final $X$ axis value than its final $Y$ axis value; this corroborates the well-known implicit bias of standard GD training. But crucially, we find that distillation *exaggerates* this bias: the student moves even faster towards its final $X$ axis value. In §5 we argue how this exaggerated bias manifests as the exaggerated confidence in Fig 1a.

However, recent work by Stanton et al. [47] has challenged our presumptions underlying what distillation supposedly does, and how it supposedly helps. First, they show in practice that the student does *not* adequately match the teacher probabilities, despite being trained to fit them. Secondly, students that do successfully match the teacher probabilities, may generalize worse than students that show some level of deviations from the teacher [47, Fig. 1]. Surprisingly, this deviation occurs even in self-distillation settings [13, 54] where the student and the teacher have identical architecture and thus the student has the potential to fit the teacher's probabilities to full precision. Even more remarkably, the self-distilled student not only deviates from the teacher's probabilities, but also supercedes the teacher in performance.

How is it possible for the student to deviate from the teacher's probabilities, and yet counter-intuitively improve its generalization beyond the teacher? Our work aims to reconcile this paradoxical behavior. On a high level, our answer is that while *arbitrary* deviations in probabilities may indeed hurt the student, in reality there are certain *systematic* deviations in the student's probabilities. Next, we argue, these systematic deviations and improved generalization co-occur because they arise from the same effect: a (helpful) form of regularization that is induced by distillation. We describe these effects in more detail in our list of key contributions below:

(i) **Exaggerated confidence**: Across a wide range of architectures and image & language classification data (spanning more than 20 settings in total) we demonstrate (§3) that *the student exaggerates the teacher's confidence*. Most typically, on low-confidence points of the teacher, the student achieves even lower confidence than the teacher; in other settings, on high-confidence points, the student achieves even higher confidence than the teacher (Fig 1a). Surprisingly, we find such deviations even with self-distillation, implying that this cannot be explained by a mere student-teacher capacity mismatch. This reveals a systematic unexpected behavior of distillation.

(ii) **Exaggerated implicit bias:** Next, we demonstrate another form of exaggeration: in some simple settings, self-distillation exaggerates the implicit bias of gradient descent (GD) in converging faster along the top data eigendirections. We demonstrate this theoretically (Thm 4.1) for linear regression, as a gradient-descent counterpart to the seminal non-gradient-descent result of Mobahi et al. [35] (see §1.1 for key differences). Empirically, we provide the first demonstration of this effect for the cross entropy loss on neural networks (Fig 1b and §4.1).

(iii) **Reconciling the paradox:** Finally, we tie the above observations together to resolve our paradox. We empirically argue how the exaggerated bias towards top eigenvectors causes the student to both (a) exaggerate confidence levels and (b) outperform the teacher (see §5). This presents a resolution for how deviations in probabilities can co-occur with improved performance.

## 1.1 Bridging key gaps between theory and practice

The resolution above helps paint a more coherent picture of theoretical and empirical studies in distillation that were otherwise disjoint. Mobahi et al. [35] proved that distillation exaggerates the bias of a *non*-gradient-descent model, one that is picked from a Hilbert space with explicit $\ell_2$ regularization. It was an open question as to whether this bias exaggeration effect is indeed relevant to settings we care about in practice. Our work establishes its relevance to practice, ultimately also drawing connections to the disjoint empirical work of Stanton et al. [47].

In more explicit terms, we establish relevance to practice in the following ways:

(1) We provide a formal proof of the exaggerated bias of distillation for a (linear) GD setting, rather than a non-GD setting (Theorem 4.1).
(2) We empirically verify the exaggerated bias of KD in more general settings e.g., a multi-layer perceptron (MLP) and a convolutional neural network (CNN) with cross-entropy loss (§4.1). This provides the first practical evidence of the bias exaggeration affect of [35].
(3) We relate the above bias to the student-teacher deviations in Stanton et al. [47]. Specifically, we argue that the exaggerated bias manifests as exaggerates student confidence levels, which we report on a wide range of image and language datasets.
(4) Tangentially, our findings also help clarify when to use early-stopping and loss-switching in distillation (§5.2).

As a more general takeaway for practitioners, our findings suggest that *not* matching the teacher probabilities exactly can be a good thing, provided the mismatch is not arbitrary. Future work may consider devising ways to explicitly induce careful deviations that further amplify the benefits of distillation e.g., by using confidence levels to reweight or scale the temperature on a per-instance basis.

## 2 Background and Notation

Our interest in this paper is *multiclass classification* problems. This involves learning a *classifier* $h: \mathcal{X} \to \mathcal{Y}$ which, for input $\mathbf{x} \in \mathcal{X}$, predicts the most likely label $h(\mathbf{x}) \in \mathcal{Y} = [K] \doteq \{1, 2, \ldots, K\}$. Such a classifier is typically implemented by computing *logits* $\mathbf{f}: \mathcal{X} \to \mathbb{R}^K$ that score the plausibility of each label, and then computing $h(\mathbf{x}) = \operatorname{argmax}_{y \in \mathcal{Y}} f_y(\mathbf{x})$. In neural models, these logits are parameterised as $\mathbf{f}(\mathbf{x}) = \mathbf{W}^\top \mathbf{Z}(\mathbf{x})$ for learned weights $\mathbf{W} \in \mathbb{R}^{D \times K}$ and embeddings $\mathbf{Z}(\mathbf{x}) \in \mathbb{R}^D$. One may learn such logits by minimising the *empirical loss* on a training sample $S \doteq \{(x_n, y_n)\}_{n=1}^N$:

$$R_{\text{emp}}(\mathbf{f}) \doteq \frac{1}{N} \sum_{n \in [N]} \mathbf{e}(y_n)^\top \ell(\mathbf{f}(\mathbf{x}_n)), \tag{1}$$

where $\mathbf{e}(y) \in \{0, 1\}^K$ denotes the *one-hot encoding* of $y$, $\ell(\cdot) \doteq [\ell(1, \cdot), \ldots, \ell(K, \cdot)] \in \mathbb{R}^K$ denotes the *loss vector* of the predicted logits, and each $\ell(y, \mathbf{f}(x))$ is the loss of predicting logits $\mathbf{f}(x) \in \mathbb{R}^K$ when the true label is $y \in [K]$. Typically, we set $\ell$ to be the softmax cross-entropy $\ell(y, \mathbf{f}(\mathbf{x})) = -\log p_y(\mathbf{x})$, where $\mathbf{p}(\mathbf{x}) \propto \exp(\mathbf{f}(\mathbf{x}))$ is the *softmax* transformation of the logits.

Equation 1 guides the learner via one-hot targets $\mathbf{e}(y_n)$ for each input. Distillation [6, 17] instead guides the learner via a target label distribution $\mathbf{p}^{\text{te}}(\mathbf{x}_n)$ provided by a *teacher*, which are the softmax probabilities from a distinct model trained on the *same* dataset. In this context, the learned model is referred to as a *student*, and the training objective is

$$R_{\text{dist}}(\mathbf{f}) \doteq \frac{1}{N} \sum_{n \in [N]} \mathbf{p}^{\text{te}}(\mathbf{x}_n)^\top \ell(\mathbf{f}(\mathbf{x}_n)). \tag{2}$$

One may also consider a weighted combination of $R_{\text{emp}}$ and $R_{\text{dist}}$, but we focus on the above objective since we are interested in understanding each objective individually.

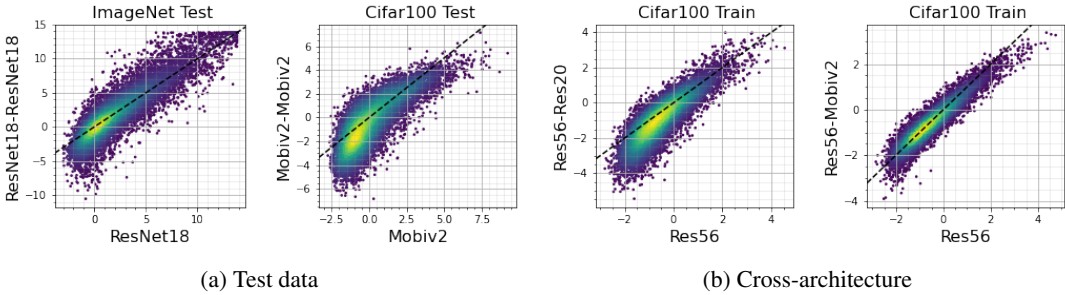

(a) Test data            (b) Cross-architecture

Figure 2: **Exaggeration of confidence in other settings.** There are settings where even on test data, and for cross-architecture distillation settings, where the student exaggerates the teacher's confidence (here specifically on low-confidence points).

Compared to training on $R_{\text{emp}}$, distillation often results in improved performance for the student [17]. Typically, the teacher model is of higher capacity than the student model; the performance gains of the student may thus informally be attributed to the teacher transferring rich information about the problem to the student. In such settings, distillation may be seen as a form of model compression. Intriguingly, however, even when the teacher and student are of the *same* capacity (a setting known as *self-distillation*), one may see gains from distillation [13, 54]. The questions we explore in this paper are motivated by the self-distillation setting; however, for a well-rounded analysis, we empirically study both the self- and cross-architecture-distillation settings.

## 3 A fine-grained look at student-teacher probability deviations

To analyze student-teacher deviations, Stanton et al. [47] measured the disagreement and the KL divergence between the student and teacher probabilities, *in expectation over all points*. They found these quantities to be non-trivially large, contrary to the premise of distillation. To probe into the exact nature of these deviations, our idea is to study the *per-sample* relationship between the teacher and student probabilities.

**Setup**. Suppose we have teacher and distilled student models $\mathbf{f}^{\text{te}}, \mathbf{f}^{\text{st}} \colon \mathcal{X} \to \mathbb{R}^K$ respectively. We seek to analyze the deviations in the corresponding predicted probability vectors $\mathbf{p}^{\text{te}}(\mathbf{x})$ and $\mathbf{p}^{\text{st}}(\mathbf{x})$ for each $(\mathbf{x}, y)$ in the train and test set, rather than in the aggregated sense as in Stanton et al. [47]. To visualize the deviations, we need a scalar summary of these vectors. An initial obvious candidate is the probabilities assigned to the ground truth class $y^\star$, namely $(p_{y^\star}^{\text{te}}(\mathbf{x}), p_{y^\star}^{\text{st}}(\mathbf{x}))$. However, the student does not have access to the ground truth class, and is only trying to mimic the teacher. Hence, it is more meaningful and valuable to focus on the *teacher's predicted class*, which the student can infer i.e., the class $y^{\text{te}} \doteq \operatorname{argmax}_{y' \in [K]} p_{y'}^{\text{te}}(\mathbf{x})$. Thus, we examine the teacher-student probabilities on this label, $(p_{y^{\text{te}}}^{\text{te}}(\mathbf{x}), p_{y^{\text{te}}}^{\text{st}}(\mathbf{x}))$. Purely for visual clarity, we further perform a monotonic logit transformation $\phi(u) = \log\left[u/(1-u)\right]$ on these probabilities to produce real values in $(-\infty, +\infty)$. Thus, we compare $\phi(p_{y^{\text{te}}}^{\text{te}}(\mathbf{x}))$ and $\phi(p_{y^{\text{te}}}^{\text{st}}(\mathbf{x}))$ for each train and test sample $(\mathbf{x}, y)$. For brevity, we refer to these values as *confidence values* for the rest of our discussion. It is worth noting that these confidence values possess another natural interpretation. For any probability vector $\mathbf{p}(\mathbf{x})$ computed from the softmax of a logit vector $\mathbf{f}(\mathbf{x})$, we can write $\phi(p_y(\mathbf{x}))$ in terms of the logits as $f_y(\mathbf{x}) - \log \sum_{k \neq y} \exp(f_k(\mathbf{x}))$. This can be interpreted as a notion of multi-class margin for class $y$.

To examine point-wise deviations of these confidence values, we consider scatter plots of $\phi(p_{y^{\text{te}}}^{\text{te}}(\mathbf{x}))$ ($X$-axis) vs. $\phi(p_{y^{\text{te}}}^{\text{st}}(\mathbf{x}))$ ($Y$-axis). We report this on the training set for some representative self-distillation settings in Figures 1a, cross-architecture distillation settings in Fig 2b and test set in Fig 2a. In all plots, the dashed line indicates the $X = Y$ line. All values are computed at the end of training. The tasks considered include image classification benchmarks, namely CIFAR10, CIFAR-100 [26], Tiny-ImageNet [27], ImageNet [45] and text classification tasks from the GLUE benchmark (e.g., MNLI [51], AGNews [55]). See §C.1 for details on the experimental hyperparameters. The plots for all the settings (spanning more than 25 settings in total) are consolidated in §C.2.

**Distilled students exaggerate one-hot trained teacher's confidence**. If the objective of distillation were minimized perfectly, the scatter plots would simply lie on $X = Y$ line. However, we find that across *all* our settings, the scatter shows stark deviations from $X = Y$. Importantly, these deviations reveal a characteristic pattern. In a vast majority of our settings, the confidence values of the student are an "exaggerated" version of the teacher's. This manifests in one of two ways. On the one hand, for points where the teacher attains low confidence, the student may attain *even lower* confidence — this is true for all our image settings, either on training or on test data. In the scatter plots, this can be inferred from the fact that for a large proportion of points where the $X$ axis value is small (low teacher confidence), it is also the case that $Y \leq X$ (even lower student confidence). As a second type of exaggeration, for points where the teacher attains high confidence, the student may attain *even higher* confidence — this is true for a majority of our language settings, barring some cross-architecture ones. To quantify these qualitative observations, in §C.4 we report the slope $m$ of the best linear fit $Y = mX + c$ over the low and high-confidence points and find that they correspond to $m > 1$ in the corresponding scenarios above.

There are two reasons why these findings are particularly surprising. First, we see these deviations in both self-distillation settings (Fig 1a) and cross-architecture settings (see Fig 2b). It is remarkable that this should occur in self-distillation given that the student has the capacity to match the teacher probabilities. Next, these deviations can occur on both training and test data (Fig 2a). Here, it is surprising that there is deviation on training data, where the student is explicitly trained to match teacher probabilities. However, we must note that there are a few exceptions where it only weakly appears in training data e.g., CIFAR-10, but in those cases it is prominent on test data.

We also conduct ablation studies in §C.5 showing that this observation is robust to various hyperparameter changes (batch size, learning rate, length of training), and choices of visualization metrics. In §C.3, we explore further patterns underlying the student's underfit points. In §C.2, we discuss the exceptions where these characteristic deviations fail to appear even if deviations are stark e.g., in cross-architecture language settings.

Thus, our point-wise visualization of deviations clarifies that the student's mismatch of the teacher's probabilities in Stanton et al. [47] stems from a systematic exaggeration of the teacher's confidence. How do we reconcile this deviation with the student outperforming the teacher in self-distillation? In the next section, we turn our attention to a different type of exaggeration exerted by distillation that will help us resolve this question.

## 4 Distillation exaggerates implicit bias of GD

While the optimal solution to the KD loss (Eq 2) is for the student to replicate the teacher's probabilities, in practice we minimize this loss using gradient descent (GD). Thus, to understand why the student exaggerates the teacher's confidence in practice, it may be key to understand how GD interacts with the distillation loss. Indeed, in this section we analyze GD and formally demonstrate that, for linear regression, distillation exaggerates a pre-existing implicit bias in GD: the tendency to converge faster along the top eigendirections of the data. We will also empirically verify that our insight generalizes to neural networks with cross-entropy loss. In a later section, we will connect this exaggeration of bias back to the exaggeration of confidence observed in §3.

Concretely, we analyze gradient flow in a linear regression setting with early-stopping. Note that linear models have been used as a way to understand distillation even in prior work [40, 35]; and early-stopping is a typical design choice in distillation practice [12, 7, 23]. Consider an $n \times p$ dataset $\mathbf{X}$ (where $n$ is the number of samples, $p$ the number of parameters) with target labels $\mathbf{y}$. Assume that the Gram matrix $\mathbf{X}\mathbf{X}^\top$ is invertible. This setting includes overparameterized scenarios ($p > n$) such as when $\mathbf{X}$ corresponds to the linearized (NTK) features of neural networks [20, 28]. Then, a standard calculation spells out the weights learned at time $t$ under GD on the loss $(1/2) \cdot \|\mathbf{X}\boldsymbol{\beta} - \mathbf{y}\|^2$ as:

$$\boldsymbol{\beta}(t) = \mathbf{X}^\top (\mathbf{X}\mathbf{X}^\top)^{-1} \mathbf{A}(t)\mathbf{y} \tag{3}$$

$$\text{where } \mathbf{A}(t) := \mathbf{I} - e^{-t\mathbf{X}\mathbf{X}^\top}. \tag{4}$$

Thus, the weights depend crucially on a time-dependent matrix $\mathbf{A}$. Intuitively, $\mathbf{A}$ determines the convergence rate independently along each eigendirection: at any time $t$, it skews the weight assigned

to an eigendirection of eigenvalue $\lambda$ by the value $(1 - e^{-\lambda t})$. As $t \to \infty$ this factor increases all the way to 1 for all directions, thus indicating full convergence. But for any finite time $t$, the topmost direction would have a larger factor than the rest, implying a bias towards that direction. Our argument is that, *distillation further exaggerates this implicit bias*. To see why, consider that the teacher is trained to time $T^{\text{te}}$. Through some calculation, the student's weights can be similarly expressed in closed-form, but with $\mathbf{A}$ replaced by the product of two matrices:

$$\tilde{\boldsymbol{\beta}}(\tilde{t}) = \mathbf{X}^\top (\mathbf{X}\mathbf{X}^\top)^{-1} \tilde{\mathbf{A}}(\tilde{t}) \mathbf{y} \tag{5}$$

$$\text{where } \tilde{\mathbf{A}}(\tilde{t}) := \mathbf{A}(t)\mathbf{A}(T^{\text{te}}). \tag{6}$$

One can then argue that the matrix $\tilde{\mathbf{A}}$ corresponding to the student is more skewed towards the top eigenvectors than the teacher:

**Theorem 4.1.** *(informal; see §B for full version and proof) Let $\beta_k(t)$ and $\tilde{\beta}_k(t)$ respectively denote the component of the teacher and student weights along the $k$'th eigenvector of the Gram matrix $\mathbf{X}\mathbf{X}^\top$, at any time $t$. Let $k_1 < k_2$ be two indices for which the eigenvalues satisfy $\lambda_{k_1} > \lambda_{k_2}$. Consider any time instants $t$ and $\tilde{t}$ at which both the teacher and the student have converged equally well along the top direction $k_1$, in that $\beta_{k_1}(t) = \tilde{\beta}_{k_1}(\tilde{t})$. Then along the bottom direction, the student has a strictly smaller component than the teacher, as in,*

$$\left| \frac{\tilde{\beta}_{k_2}(\tilde{t})}{\beta_{k_2}(t)} \right| < 1. \tag{7}$$

The result says that the student relies less on the bottom eigendirections than the teacher, if we compare them at any instant when they have both converged equally well along the top eigendirections. In other words, while the teacher already has an implicit tendency to converge faster along the top eigendirections, the student has an even stronger tendency to do so. In the next section, we demonstrate that this insight generalizes to more practical non-linear settings.

**Connection to prior theory.** As discussed earlier, we build on Mobahi et al. [35] who prove that distillation exaggerates the *explicit regularization* applied in a *non-GD* setting. However, it was an open question as to whether their insight is relevant to GD-trained models used in practice, which we answer in the affirmative. We further directly establish its relevance to practice through an empirical demonstration of our insights in more general neural network settings in §4.1. Also note that linear models were studied by Phuong and Lampert [40] too, but they do not show how the student learns any different weights than the teacher, let alone *better* weights than the teacher.

Our result also brings out an important clarification regarding early-stopping. Mobahi et al. [35] argue that early-stopping and distillation have opposite regularization effects, wherein the former has a densifying effect while the latter a sparsifying effect. However, this holds only in their non-GD setting. In GD settings, we argue that distillation amplifies the effect of early stopping, rather than oppose it.

## 4.1 Empirical verification of exaggerated bias in more general settings

While our theory applies to linear regression with gradient *flow*, we now verify our insights in more general settings. In short, we consider settings with (a) finite learning rate instead of infinitesimal, (b) cross-entropy instead of squared error loss, (c) an MLP (in §D, we consider a CNN), (d) trained on a non-synthetic dataset (MNIST).

To examine how the weights evolve in the eigenspace, we project the first layer weights $\mathbf{W}$ onto each eigenvector of the data $\mathbf{v}$ to compute the component $\|\mathbf{W}^\top \mathbf{v}\|_2$. We then sample two eigenvectors $\mathbf{v}_i, \mathbf{v}_j$ at random (with $i < j$), and plot how the quantities $(\|\mathbf{W}^\top \mathbf{v}_i\|_2, \|\mathbf{W}^\top \mathbf{v}_j\|_2)$ evolve over time. This provides a 2-D glimpse into a complex high-dimensional trajectory of the model.

We provide two such random 2-D slices of the trajectory for our MLP in Fig 1b, and many more such random slices in §D (not cherry-picked). Across almost all of these slices, we find a consistent pattern emerge. First, as is expected, the one-hot trained teacher shows an implicit bias towards converging to its final value faster along the top direction, which is plotted along the $X$ axis. But crucially, across almost all these slices, the distilled student presents a more exaggerated version of this bias. This leads it to traverse *a different* part of the parameter space with greater reliance on the

top directions. Notably, we also find underfitting of low-confidence points in this setting, visualized in §D.

## 5 Reconciling student-teacher deviations and generalization

So far, we have established two forms of exaggeration under distillation, one of confidence (§3), and one of convergence in the eigenspace (§4). Next, via a series of experiments, we tie these together to resolve our core paradox: that the student-teacher deviations somehow co-occur with the improved performance of the student. First, we design an experiment demonstrating how the exaggerated bias from §4 translates into exaggerated confidence levels seen in the probability deviation plots of §3. Then, via some controlled experiments, we argue when and how this bias can also simultaneously lead to improved student performance. Thus, our overall argument is that both the deviations and the improved performance *can* co-occur thanks to a common factor, the exaggeration in bias. We summarize this narrative as a graph in Fig 3.

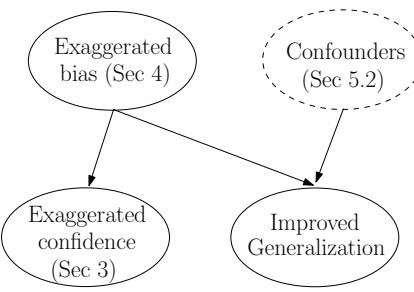

Figure 3: **Reconciling the paradox:** Distillation exaggerates the implicit bias of GD, which can both exaggerate confidence levels (thus causing deviations in probability) and help generalization. Note that the improved generalization is however conditioned on other confounding factors such as the teacher's training accuracy, as we discuss later in § 5.2.

### 5.1 Connecting exaggerated bias to deviations and generalization

**Exaggerated implicit bias results in exaggerated confidence.** We frame our argument in an empirical setting where a portion of the CIFAR100 one-hot labels are mislabeled. In this context, it is well-known that the implicit bias of early-stopped GD fits the noisy subset of the data more slowly than the clean data; this is because noisy labels correspond to bottom eigendirections [29, 12, 4, 24]. Even prior works on distillation [12, 43] have argued that when observed labels are noisy, the teacher's implicit bias helps denoise the labels. As a first step of our argument, we corroborate this in Fig 4, where we indeed find that mislabeled points have the low teacher confidence (small $X$ axis values).

Going beyond this prior understanding, we make a crucial second step in our argument: our theory would predict that the student must rely *even less* on the lower eigendirections than the teacher already does. This means an even poorer fit of the mislabeled datapoints than the teacher. Indeed, in Fig 4, we find that of all the points that the teacher has low confidence on (i.e., points with small $X$ values) — which includes some clean data as well — the student underfits all the *mis*labeled data (i.e., $Y < X$ in the plots for those points). This confirms our hypothesis that the exaggeration of the implicit bias in the eigenspace translates to an exaggeration of the confidence levels, and thus a deviation in probabilities.

**Exaggerated implicit bias can result in student outperforming the teacher.** The same experiment above also gives us a handle on understanding how distillation benefits generalization. Specifically, in Table 2, we find that in this setting, the self-distilled ResNet56 model witnesses a $3\%$ gain over an identical teacher. Prior works [12, 43] argue, this is because the implicit bias of the teacher results in teacher probabilities that are partially denoised compared to the one hot labels. This alone however, cannot explain why the student — which is supposedly trying to replicate the teacher's probabilities — can *outperform* the teacher. Our solution to this is to recognize that the student doesn't replicate the teacher's bias towards top eigenvectors, but rather exaggerates it. This provides an enhanced denoising which is crucial to outperforming the teacher. The exaggerated bias not only helps generalization, but as discussed in the previous paragraph, it also induces deviations in probability. This provides a reconciliation between the two seemingly inconsistent behaviors.

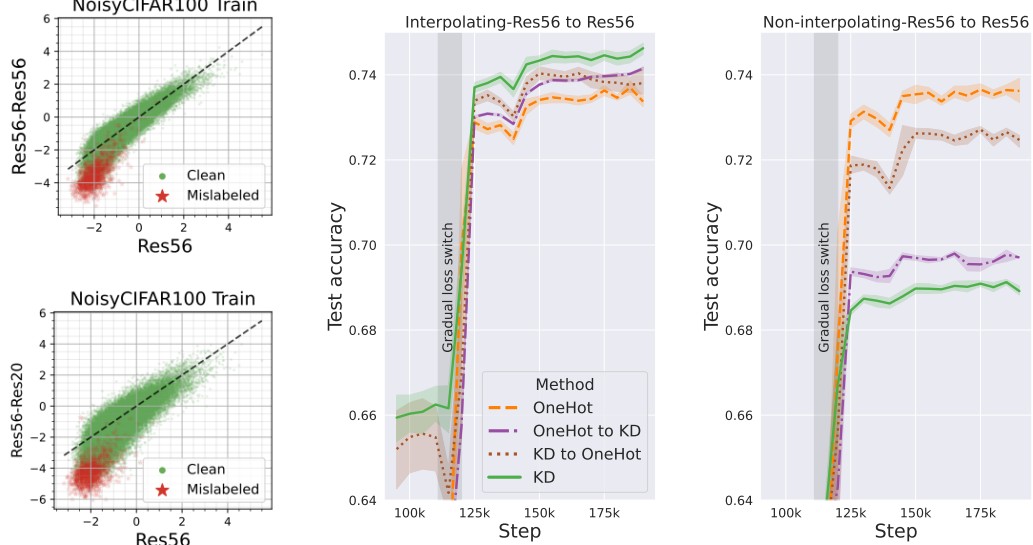

(a) Confidence exaggeration

(b) Effect of teacher interpolation level on CIFAR-100 self-distillation

Figure 4: **Left: Exaggeration of confidence under explicit label noise:** While the teacher already achieves low confidence on points with wrong one-hot labels, the student achieves even lower confidence on these points, in both self- (top) and cross-architecture (bottom) distillation. **Right: Effect of teacher's interpolation level in CIFAR-100:** For an interpolating teacher (left), switching to one-hot loss in the middle of training *hurts* generalization, while for a non-interpolating teacher, the switch to one-hot is helpful.

## 5.2 When distillation can hurt generalization

We emphasize an important nuance to this discussion: regularization can also hurt generalization, if other confounding factors (e.g., dataset complexity) are unfavorable. Below, we discuss a key such confounding factor relevant to our experiments.

**Teacher's top-1 train accuracy as a confounding factor.** A well-known example of where distillation hurts generalization is that of ImageNet, as demonstrated in Fig. 3 of Cho and Hariharan [7]. We corroborate this in our setup in Table 2. At the same time, in our experiments on ImageNet, we find that distillation does exaggerate the confidence levels (Fig 7), implying that regularization is at play. A possible explanation for why the student suffers here could be that it has inadequate capacity to match the rich non-target probabilities of the teacher [7, 21]. However, this cannot justify why even self-distillation is detrimental in ImageNet (e.g., [7, Table 3] for ResNet18 self-distillation).

We advocate for an alternative hypothesis, in line with [19, 58]: *distillation can hurt the student when the teacher does not achieve sufficient top-1 accuracy on the training data.* e.g., ResNet18 has 78% ImageNet train accuracy. This hypothesis may appear to contradict the observation from [7] that the student's accuracy is hurt by much larger teachers, which have better training accuracy. However, in the presence of a larger teacher, there are two confounding factors: the teacher's train accuracy and the complexity of the teacher's non-target probabilities. This makes it hard to disentangle the individual effect of the two factors, which we claim, have opposite effects on the student's performance.

To isolate the effect of the teacher's top-1 training accuracy, we focus on the self-distillation setting. In this setting, we provide three arguments supporting the hypothesis that the teacher's imperfect training accuracy can hurt the student.

**Evidence 1: A controlled experiment with an interpolating and a non-interpolating teacher.** First, we train two ResNet56 teachers on CIFAR100, one which interpolates on the whole dataset (i.e., 100% top-1 accuracy), and another which interpolates on only half the dataset. Upon distilling a ResNet56 student on the *whole* dataset in both settings, we find in Fig 4b that distilling from the

interpolating teacher helps, while distilling from the non-interpolating teacher hurts. This provides direct evidence for our argument.

**Evidence 2: Switching to one-hot loss *helps* under a *non*-interpolating teacher.** For a non-interpolating teacher, distillation must provide rich top-K information while one-hot must provide precise top-1 information. Thus, our hypothesis would predict that for a non-interpolating teacher, there must be a way to optimally train the student with both distillation and one-hot losses. Indeed [7, 21, 58] already demonstrate that making some sort of soft switch from distillation to one-hot loss over the course of training, improves generalization for ImageNet. Although [7] motivate this from their capacity mismatch hypothesis, they report that this technique works for self-distillation on ImageNet as well (e.g., [7, Table 3]), thus validating our hypothesis. We additionally verify these findings indeed hold in some of our self-distillation settings, namely the (controlled) non-interpolating CIFAR100 teacher (Fig 4b), and a (naturally) non-interpolating TinyImagenet teacher (§E), where the capacity mismatch argument does not apply.

**Evidence 3: Switching to one-hot loss *hurts* under an interpolating teacher.** Our hypothesis would predict that a switch from distillation to one-hot loss, would not be helpful if the teacher already has perfect top-1 accuracy. We verify this with a interpolating CIFAR100 teacher (Fig 4b, Fig 24). Presumably, one-hot labels provide strictly less information in this case, and causes the network to overfit to the less informative signals. This further reinforces the hypothesis that the teacher's top-1 training accuracy is an important factor in determining whether the exaggerated bias effect of distillation helps generalization.

Framed within the terms of our eigenspace analysis, when the teacher has imperfect top-1 training accuracy, it may mean that the teacher has not sufficiently converged along some critical (say, second or third) eigendirections of the data. The bias exaggerated by distillation would only further curtail the convergence along these directions, hurting generalization.

In summary, this discussion leads us to a more nuanced resolution to the apparent paradox of student-teacher deviations co-occuring with improved generalization. On the one hand, distillation causes an exaggeration of the confidence levels, which causes a deviation between student and teacher probabilities. At the same time, the same effect can aid the student's generalization, *provided other confounding factors are conducive for it.*

# 6    Relation to Existing Work

**Distillation as a probability matching process.** Distillation has been touted to be a process that benefits from matching the teacher's probabilities [17]. Indeed, many distillation algorithms have been designed in a way to more aggressively match the student and teacher functions [9, 5]. Theoretical analyses too rely on explaining the benefits of distillation based on a student that obediently matches the teacher's probabilities [34]. But, building on Stanton et al. [47], our work demonstrates why we may desire that the student deviate from the teacher, in certain systematic ways.

**Theories of distillation.** A long-standing intuition for why distillation helps is that the teacher's probabilities contain "dark knowledge" about class similarities [17, 36], Several works [34, 10, 43, 57] have formalized these similarities via inherently noisy class memberships. However, some works [13, 53, 48] have argued that this hypothesis cannot be the sole explanation, because distillation can help even if the student is only taught information about the target probabilities (e.g., by smoothing out all non-target probabilities).

This has resulted in various alternative hypotheses. Some have proposed faster convergence [40, 42, 23] which only explains why the student would converge fast to the teacher, but not why it may deviate from and supersede a one-hot teacher. Another line of work casts distillation as a regularizer, either in the sense of Mobahi et al. [35] or in the sense of instance-specific label smoothing [56, 53, 48]. Another hypothesis is that distillation induces better feature learning or conditioning [2, 22], likely in the early parts of training. This effect however is not general enough to appear in convex linear settings, where distillation can help. Furthermore, it is unclear if this is relevant in the CIFAR100 setting, where we find that switching to KD much later during training is sufficient to see gains in distillation (§E). Orthogonally, [39] suggest that distillation results in flatter minima, which may lead to better generalization. Finally, we also refer the reader to [32, 25, 50] who theoretically study distillation in orthogonal settings.

**Early-stopping and knowledge distillation.** Early-stopping has received much attention in the context of distillation [30, 43, 12, 7, 23]. We build on Dong et al. [12], who argue how early-stopping a GD-trained teacher can automatically denoise the labels due to regularization in the eigenspace. However, these works do not provide an argument for why distillation can outperform the teacher.

**Empirical studies of distillation.** Our study crucially builds on observations from [47, 33] demonstrating student-teacher deviations in an aggregated sense than in a sample-wise sense. Other studies [1, 37] investigate how the student is *similar* to the teacher in terms of out-of-distribution behavior, calibration, and so on. Deng and Zhang [11] show how a smaller student can outperform the teacher when allowed to match the teacher on more data, which is orthogonal to our setting.

## 7    Discussion and Future Work

Here, we highlight the key insights from our work valuable for future research in distillation practice:

1. *Not matching the teacher probabilities exactly can be a good thing, if done carefully*. Perhaps encouraging underfitting of teachers' low-confidence points can further exaggerate the benefits of the regularization effect.
2. *It may help to switch to one-hot loss in the middle of training if the teacher does not sufficiently interpolate the ground truth labels.*

We also highlight a few theoretical directions for future work. First, it would be valuable to extend our eigenspace view to multi-layered models where the eigenspace regularization effect may "compound" across layers. Furthermore, one could explore ways to exaggerate the regularization effect in our simple linear setting and then extend the idea to a more general distillation approach. Finally, it would be practically useful to extend these insights to other modes of distillation, such as *semi-supervised* distillation [8], non-classification settings such as ranking models [18], or intermediate-layer-based distillation [44].

We highlight the limitations of our study in Appendix A.

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

# Appendix

## Table of Contents

## A Limitations

We highlight a few key limitations to our results that may be relevant for future work to look at:

1. Our visualizations focus on student-teacher deviations in the top-1 class of the teacher. While this already reveals a systematic pattern across various datasets, this does not capture richer deviations that may occur in the teacher's lower-ranked classes. Examining those would shed light on the "dark knowledge" hidden in the non-target classes.

2. Although we demonstrate the exaggerated bias of Theorem 4.1 in MLPs (Sec D, Fig 20) and CNNs (Sec D, Fig 21), we do not formalize any higher-order effects that may emerge in such multi-layer models. It is possible that the same eigenspace regularization effect propagates down the layers of a network. We show some preliminary evidence in Sec D.7.

3. We do not exhaustively characterize when the underlying exaggerated bias of distillation is *(in)sufficient* for improved generalization. One example where this relationship is arguably sufficient is in the case of noise in the one-hot labels (Fig 4). One example where this is insufficient is when the teacher does not fit the one-hot labels perfectly (Fig 4b). A more exhaustive characterization would be practically helpful as it may help us predict when it is worth performing distillation.

4. The effect of the teacher's top-1 accuracy (Sec 5.2) has a further confounding factor which we do not address: the "complexity" of the dataset. For CIFAR-100, the teacher's labels are more helpful than the one-hot labels, even for a mildly-non-interpolating teacher with $4\%$ top-1 error on training data; it is only when there is sufficient lack of interpolation that one-hot labels complement the teacher's labels. For the relatively more complex Tiny-Imagenet, the one-hot labels complement teacher's soft labels even when the teacher has $2\%$ top-1 error (Fig 24).

## B Proof of Theorem

Below, we provide the proof for Theorem 4.1 that shows that the distilled student converges faster along the top eigendirections than the teacher.

**Theorem B.1.** *Let $\mathbf{X} \in \mathbb{R}^{n \times p}$ and $\mathbf{y} \in \mathbb{R}^n$ be the p-dimenionsional inputs and labels of a dataset of $n$ examples, where $p > n$. Assume the Gram matrix $\mathbf{X}\mathbf{X}^\top$ is invertible, with $n$ eigenvectors $\mathbf{v}_1, \mathbf{v}_2, \ldots, \mathbf{v}_n$ in $\mathbb{R}^p$. Let $\boldsymbol{\beta}(t) \in \mathbb{R}^p$ denote a teacher model at time $t$, when trained with gradient flow to minimize $\frac{1}{2}\|\mathbf{X}\boldsymbol{\beta}(t) - \mathbf{y}\|^2$, starting from $\boldsymbol{\beta}(0) = \mathbf{0}$. Let $\tilde{\boldsymbol{\beta}}(\tilde{t}) \in \mathbb{R}^p$ be a student model at time $\tilde{t}$, when trained with gradient flow to minimize $\frac{1}{2}\|\mathbf{X}\boldsymbol{\beta}(t) - \mathbf{y}^{te}\|^2$, starting from $\tilde{\boldsymbol{\beta}}(0) = \mathbf{0}$; here $\mathbf{y}^{te} = \mathbf{X}\boldsymbol{\beta}(T^{te})$ is the output of a teacher trained to time $T^{te} > 0$. Let $\beta_k(\cdot)$ and $\tilde{\beta}_k(\cdot)$ respectively denote the component of the teacher and student weights along the $k$'th eigenvector of the Gram matrix $\mathbf{X}\mathbf{X}^\top$ as:*

$$\beta_k(t) = \boldsymbol{\beta}_k(t) \cdot \mathbf{v}_k, \tag{8}$$

*and*

$$\tilde{\beta}_k(\tilde{t}) = \tilde{\boldsymbol{\beta}}_k(\tilde{t}) \cdot \mathbf{v}_k. \tag{9}$$

*Let $k_1 < k_2$ be two indices for which the eigenvalues satisfy $\lambda_{k_1} > \lambda_{k_2}$, if any exist. Consider any time instants $t > 0$ and $\tilde{t} > 0$ at which both the teacher and the student have converged equally well along the top direction $\mathbf{v}_{k_1}$, in that*

$$\beta_{k_1}(t) = \tilde{\beta}_{k_1}(\tilde{t}). \tag{10}$$

*Then along the bottom direction, the student has a strictly smaller component than the teacher, as in,*

$$\left| \frac{\tilde{\beta}_{k_2}(\tilde{t})}{\beta_{k_2}(t)} \right| < 1. \tag{11}$$

*Proof.* (of Theorem 4.1)

Recall that the closed form solution for the teacher is given as:

$$\boldsymbol{\beta}(t) = \mathbf{X}^\top (\mathbf{X}\mathbf{X}^\top)^{-1} \mathbf{A}(t)\mathbf{y} \tag{12}$$

$$\text{where } \mathbf{A}(t) := \mathbf{I} - e^{-t\mathbf{X}\mathbf{X}^\top}. \tag{13}$$

Similarly, by plugging in the teacher's labels into the above equation, the closed form solution for the student can be expressed as:

$$\tilde{\boldsymbol{\beta}}(\tilde{t}) = \mathbf{X}^\top (\mathbf{X}\mathbf{X}^\top)^{-1} \tilde{\mathbf{A}}(\tilde{t})\mathbf{y} \tag{14}$$

$$\text{where } \tilde{\mathbf{A}}(\tilde{t}) := \mathbf{A}(t)\mathbf{A}(T^{\text{te}}). \tag{15}$$

Let $\alpha_k(t), \tilde{\alpha}_k(\tilde{t})$ be the eigenvalues of the $k$'th eigendirection in $\mathbf{A}(t)$ and $\tilde{\mathbf{A}}(\tilde{t})$ respectively. We are given $\beta_{k_1}(t) = \tilde{\beta}_{k_1}(\tilde{t})$. From the closed form expression for the two models in Eq 12 and Eq 14, we can infer $\alpha_{k_1}(t) = \tilde{\alpha}_{k_1}(\tilde{t})$. Similarly, from the closed form expression, it follows that in order to prove $|\beta_{k_2}(t)| > |\tilde{\beta}_{k_2}(\tilde{t})|$, it suffices to prove $\alpha_{k_2}(t) > \tilde{\alpha}_{k_2}(\tilde{t})$.

For the rest of the discussion, for convenience of notation, we assume $k_1 = 1$ and $k_2 = 2$ without loss of generality. Furthermore, we define $\alpha_1^\star = \alpha_1(t) = \tilde{\alpha}_1(\tilde{t})$.

From the teacher's system of equations in Eq 13, $\alpha_1^\star = 1 - e^{-\lambda_1 t}$. Hence, we can re-write $\alpha_2(t)$ as:

$$\alpha_2(t) = 1 - e^{-\lambda_2 t} \tag{16}$$

$$= 1 - \left(e^{-\lambda_1 t}\right)^{\frac{\lambda_2}{\lambda_1}} \tag{17}$$

$$= 1 - (1 - \alpha_1^\star)^{\frac{\lambda_2}{\lambda_1}}. \tag{18}$$

Similarly for the student, from Eq 15,

$$\alpha_1^\star = (1 - e^{-\lambda_1 \tilde{t}})(1 - e^{-\lambda_1 T^{\text{te}}}). \tag{19}$$

Hence, we can re-write $\tilde{\alpha}_2(\tilde{t})$ as:

$$\tilde{\alpha}_2(\tilde{t}) = (1 - e^{-\lambda_2 \tilde{t}}) \cdot (1 - e^{-\lambda_2 T^{\text{te}}}) \tag{20}$$

$$= \left(1 - \left(e^{-\lambda_1 \tilde{t}}\right)^{\frac{\lambda_2}{\lambda_1}}\right) \cdot \left(1 - \left(e^{-\lambda_1 T^{\text{te}}}\right)^{\frac{\lambda_2}{\lambda_1}}\right) \tag{21}$$

For convenience, let us define $a := e^{-\lambda_1 \tilde{t}}$, $b := e^{-\lambda_1 T^{\text{te}}}$ and $\kappa = \lambda_2/\lambda_1$. Then, rewriting Eq 19, we get

$$\alpha_1^\star = (1 - a)(1 - b). \tag{22}$$

Plugging this into Eq 18,

$$\alpha_2(t) = 1 - (1 - (1 - a)(1 - b))^\kappa. \tag{23}$$

Similarly, rewriting Eq 21, in terms of $a, b, \kappa$:

$$\tilde{\alpha}_2(\tilde{t}) = (1 - a^\kappa)(1 - b^\kappa). \tag{24}$$

We are interested in the sign of $\alpha_2(t) - \tilde{\alpha}_2(\tilde{t})$. Let $f(u) = u^\kappa + (a + b - u)^\kappa$. Then, we can write this difference as follows:

$$\alpha_2(t) - \tilde{\alpha}_2(\tilde{t}) = a^\kappa + b^\kappa - (ab)^\kappa - (1 - (1 - a)(1 - b))^\kappa \tag{25}$$

$$= a^\kappa + b^\kappa - ((ab)^\kappa + (a + b - ab)^\kappa) \tag{26}$$

$$= f(a) - f(a + b(1 - a)) = f(b) - f(b + a(1 - b)). \tag{27}$$

To prove that last expression in terms of $f$ resolves to a positive value, we make use of the fact that when $\kappa \in (0, 1)$, $f(u)$ attains its maximum at $u = \frac{a+b}{2}$, and is monotonically decreasing for $u \in \left[\frac{a+b}{2}, a + b\right]$. Note that $\kappa$ is indeed in $(0, 1)$ because $\lambda_2 < \lambda_1$. Since $\tilde{t} > 0$ and $T^{\text{te}} > 0$, $a \in (0, 1)$ and $b \in (0, 1)$. Since $f$ is symmetric with respect to $a$ and $b$, without loss of generality, let $a$ be the larger of $\{a, b\}$.

Since $a < 1$, and $b > 0$, we have $a + b(1 - a) > a$. Also since $a$ is the larger of the two, we have $a > \frac{a+b}{2}$. Combining these two, $a + b > a + b(1 - a) > a > \frac{a+b}{2}$. Thus, from the monotonic decrease of $f$ for $u \in \left[\frac{a+b}{2}, a + b\right]$, $f(a) > f(a + b(1 - a))$. Thus,

$$\alpha_2(t) - \tilde{\alpha}_2(\tilde{t}) > 0, \tag{28}$$

proving our claim.

$\square$

Table 1: Summary of training settings on image data.

| Hyperparameter (based on) | CIFAR10* v1 | CIFAR100 v2 Tian et al. [49] | Tiny-ImageNet | ImageNet Cho and Hariharan [7] |
|---|---|---|---|---|
| Weight decay | $5 \cdot 10^{-4}$ | $5 \cdot 10^{-4}$ | $5 \cdot 10^{-4}$ | $10^{-4}$ |
| Batch size | 1024 | 64 | 128 | 1024 |
| Epochs | 450 | 240 | 200 | 90 |
| Peak learning rate | 1.0 | 0.05 | 0.1 | 0.4 |
| Learning rate warmup epochs | 15 | 1 | 5 | 5 |
| Learning rate decay factor | 0.1 | 0.1 | 0.1 | Cosine schedule |
| Nesterov momentum | 0.9 | 0.9 | 0.9 | 0.9 |
| Distillation weight | 1.0 | 1.0 | 1.0 | 0.1 |
| Distillation temperature | 4.0 | 4.0 | 4.0 | 4.0 |
| Gradual loss switch window | $1k$ steps | $1k$ steps | $10k$ steps | $1k$ steps |

## C  Further experiments on student-teacher deviations

### C.1  Details of experimental setup

We present details on relevant hyper-parameters for our experiments.

**Model architectures**. For all image datasets (CIFAR10, CIFAR100, Tiny-ImageNet, ImageNet), we use ResNet-v2 [15] and MobileNet-v2 [46], models. Specifically, for CIFAR, we consider the CIFAR ResNet-$\{56, 20\}$ family and MobileNet-v2 architectures; for Tiny-ImageNet, we consider the ResNet-$\{50, 18\}$ family and MobileNet-v2 architectures; for ImageNet we consider ResNet-18 family based on the TorchVision implementation. For all ResNet models, we employ standard augmentations as per He et al. [16].

For all text datasets (MNLI, AGNews, QQP, IMDB), we fine-tune a pre-trained RoBERTa [31] model. We consider combinations of cross-architecture- and self-distillation with RoBERTa -Base, -Medium and -Small architectures.

**Training settings**. We train using minibatch SGD applied to the softmax cross-entropy loss. For all image datasets, we follow the settings in Table 1. For the noisy CIFAR dataset, for $20\%$ of the data we randomly flip the one-hot label to another class. Also note that, we explore two different hyperparameter settings for CIFAR100, for ablation. For all text datasets, we use a batch size of 64, and train for 25000 steps. We use a peak learning rate of $10^{-5}$, with 1000 warmup steps, decayed linearly. For the distillation experiments on text data, we use a distillation weight of 1.0. We use temperature $\tau = 2.0$ for MNLI, $\tau = 16.0$ for IMDB, $\tau = 1.0$ for QQP, and $\tau = 1.0$ for AGNews.

For all CIFAR experiments in this section we use GPUs. These experiments take a couple of hours. We run all the other experiments on TPUv3. The ImageNet experiments take around 6-8 hours, TinyImagenet a couple of hours and the RoBERTA-based experiments take $\approx 12$ hours. Note that for all the later experiments in support of our eigenspace theory (Sec D), we only use a CPU; these finish in few minutes each.

### C.2  Scatter plots of probabilities

In this section, we present additional scatter plots of the teacher-student logit-transformed probabilities for the class corresponding to the teacher's top prediction: Fig 7 (for ImageNet), Fig 5,6 (for CIFAR100), Fig 8 (for TinyImagenet), Fig 9 (for CIFAR10), Fig 10 (for MNLI and AGNews settings), Fig 11 (for further self-distillation on QQP, IMDB and AGNews) and Fig 12 (for cross-architecture distillation on language datasets). Below, we qualitatively describe how confidence exaggeration manifests (or does not) in these settings. We attempt a quantitative summary subsequently in Sec C.4.

**Image data.** First, across *all* the 18 image settings, we observe an underfitting of the low-confidence points on *test* data. Note that this is highly prominent in some settings (e.g., CIFAR100, MobileNet self-distillation in Fig 5 fourth column, second row), but also faint in other settings (e.g., CIFAR100, ResNet56-ResNet20 distillation in Fig 5 second column, second row).

Table 2: Summary of train and test performance of various distillation settings.

| Dataset | Teacher | Student | Train accuracy | | | Test accuracy | | |
|---|---|---|---|---|---|---|---|---|
| | | | Teacher | Student (OH) | Student (DIST) | Teacher | Student (OH) | Student (DIST) |
| CIFAR10 | ResNet-56 | ResNet-56 | 100.00 | 100.00 | 100.00 | 93.72 | 93.72 | 93.9 |
| | ResNet-56 | ResNet-20 | 100.00 | 99.95 | 99.60 | 93.72 | 91.83 | 92.94 |
| | ResNet-56 | MobileNet-v2-1.0 | 100.00 | 100.00 | 99.96 | 93.72 | 85.11 | 87.81 |
| | MobileNet-v2-1.0 | MobileNet-v2-1.0 | 100.00 | 100.00 | 100.00 | 85.11 | 85.11 | 86.76 |
| CIFAR100 | ResNet-56 | ResNet-56 | 99.97 | 99.97 | 97.01 | 72.52 | 72.52 | 74.55 |
| | ResNet-56 | ResNet-20 | 99.97 | 94.31 | 84.48 | 72.52 | 67.52 | 70.87 |
| | MobileNet-v2-1.0 | MobileNet-v2-1.0 | 99.97 | 99.97 | 99.96 | 54.32 | 54.32 | 56.32 |
| | ResNet-56 | MobileNet-v2-1.0 | 99.97 | 99.97 | 99.56 | 72.52 | 54.32 | 62.40 |
| (v2 hyperparams.) | ResNet-56 | ResNet-56 | 96.40 | 96.40 | 87.61 | 73.62 | 73.62 | 74.40 |
| CIFAR100 (noisy) | ResNet-56 | ResNet-56 | 99.9 | 99.9 | 95.6 | 69.8 | 69.8 | 72.7 |
| | ResNet-56 | ResNet-20 | 99.9 | 91.4 | 82.8 | 69.8 | 64.9 | 69.2 |
| Tiny-ImageNet | ResNet-50 | ResNet-50 | 98.62 | 98.62 | 94.84 | 66 | 66 | 66.44 |
| | ResNet-50 | ResNet-18 | 98.62 | 93.51 | 91.09 | 66 | 62.78 | 63.98 |
| | ResNet-50 | MobileNet-v2-1.0 | 98.62 | 89.34 | 87.90 | 66 | 62.75 | 63.97 |
| | MobileNet-v2-1.0 | MobileNet-v2-1.0 | 89.34 | 89.34 | 82.26 | 62.75 | 62.75 | 63.28 |
| ImageNet | ResNet-18 | ResNet-18 (full KD) | 78.0 | 78.0 | 72.90 | 69.35 | 69.35 | 69.35 |
| | ResNet-18 | ResNet-18 (late KD) | 78.0 | 78.0 | 71.65 | 69.35 | 69.35 | 68.3 |
| | ResNet-18 | ResNet-18 (early KD) | 78.0 | 78.0 | 79.1 | 69.35 | 69.35 | 69.75 |
| MNLI | RoBERTa-Base | RoBERTa-Small | 92.9 | 72.1 | 72.6 | 87.4 | 69.9 | 70.3 |
| | RoBERTa-Base | RoBERTa-Medium | 92.9 | 88.2 | 86.8 | 87.4 | 83.8 | 84.1 |
| | RoBERTa-Small | RoBERTa-Small | 72.1 | 72.1 | 71.0 | 69.9 | 69.9 | 69.9 |
| | RoBERTa-Medium | RoBERTa-Medium | 88.2 | 88.2 | 85.6 | 83.8 | 83.8 | 83.5 |
| IMDB | RoBERTa-Small | RoBERTa-Small | 100.0 | 100.0 | 99.1 | 90.4 | 90.4 | 91.0 |
| | RoBERTa-Base | RoBERTa-Small | 100.0 | 100.0 | 99.9 | 95.9 | 90.4 | 90.5 |
| QQP | RoBERTa-Small | RoBERTa-Small | 85.0 | 85.0 | 83.2 | 83.5 | 83.5 | 82.5 |
| | RoBERTa-Medium | RoBERTa-Medium | 92.3 | 92.3 | 90.5 | 89.7 | 89.7 | 89.0 |
| | RoBERTa-Base | RoBERTa-Small | 93.5 | 85.0 | 85.1 | 90.5 | 83.5 | 84.0 |
| AGNews | RoBERTa-Small | RoBERTa-Small | 96.3 | 96.3 | 95.7 | 93.6 | 93.6 | 93.3 |
| | RoBERTa-Base | RoBERTa-Medium | 99.2 | 98.4 | 97.8 | 95.2 | 95.2 | 94.5 |
| | RoBERTa-Base | RoBERTa-Small | 99.2 | 96.3 | 96.0 | 95.2 | 93.6 | 93.6 |

Second, on the training data, this occurs in a majority of settings (13 out of 18) except CIFAR100 MobileNet self-distillation (Fig 5 fourth column) and three of the four CIFAR10 experiments. In all the CIFAR100 settings where this occurs, this is more prominent on training data than on test data.

Third, in a few settings, we also find an overfitting of high-confidence points, indicating a second type of exaggeration. In particular, this occurs for our second hyperparameter setting in CIFAR100 (Fig 6 last column), Tiny-ImageNet with a ResNet student (Fig 8 first and last column).

**Language data.** In the language datasets, we find the student-teacher deviations to be different in pattern from the image datasets. We find for lower-confidence points, there is typically both significant underfitting and overfitting (i.e., $|Y - X|$ is larger for small $X$); for high-confidence points, there is less deviation, and if any, the deviation is from overfitting ($Y > X$ for large $X$).

This behavior is most prominent in four of the settings plotted in Fig 10. We find a weaker manifestation in four other settings in Fig 11. Finally in Fig 12, we report the scenarios where we do not find a meaningful behavior. What is however consistent is that there is always a stark deviation in all the above settings.

**Exceptions:** In summary, we find patterns in all but the following exceptions:

1. For MobileNet self-distillation on CIFAR100, and for three of the CIFAR10 experiments, we find no underfitting of the lower-confidence points *on the training dataset* (but they hold on test set). Furthermore, in all these four settings, we curiously find an underfitting of the high-confidence points in both test and training data.

2. Our patterns break down in a four of the *cross-architecture* settings of language datasets. This may be because certain cross-architecture effects dominate over the more subtle underfitting effect.

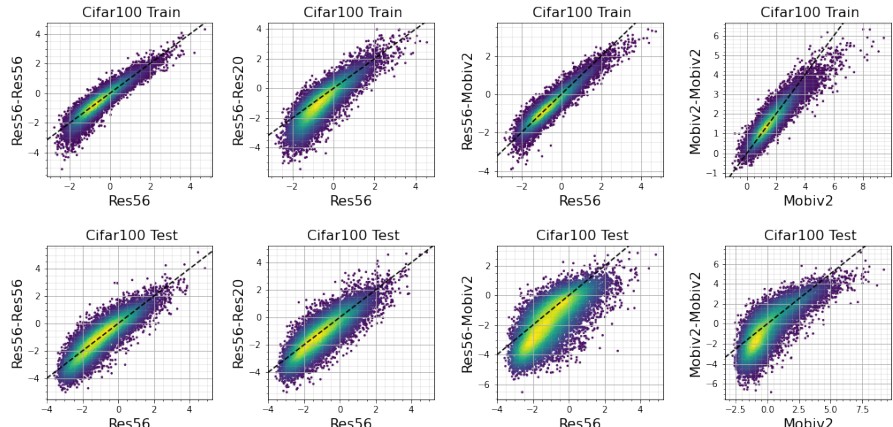

Figure 5: **Teacher-student logit plots for CIFAR100 experiments:** We report plots for various distillation settings involving ResNet56, ResNet20 and MobileNet-v2 (training data on top, test data in the bottom). We find underfitting of the low-confidence points in the training set in all but the MobileNet self-distillation setting. But in all the settings, we find significant underfitting of the low-confidence points in the *test* dataset.

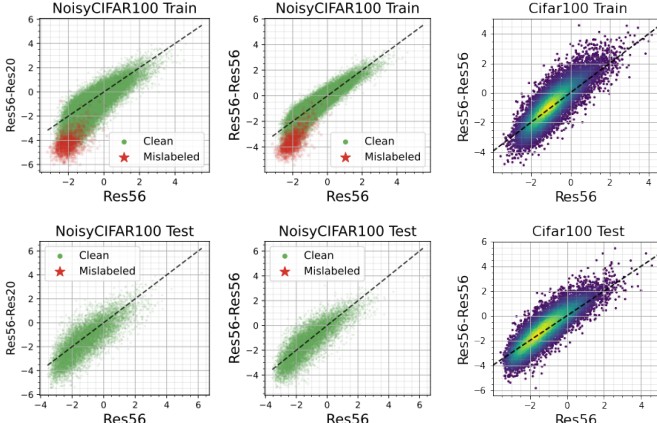

Figure 6: **Teacher-student logit plots for more CIFAR100 experiments:** We report underfitting of low-confidence points for a few other CIFAR100 distillation settings. The first column is self-distillation setting where 20% of one-hot labels are noisy; the second column on the same data, but cross-architecture; the last column is ResNet-56 self-distillation on the original CIFAR100, but with another set of hyperparameters specified in Table 1. Here we also find overfitting of high-confidence points.

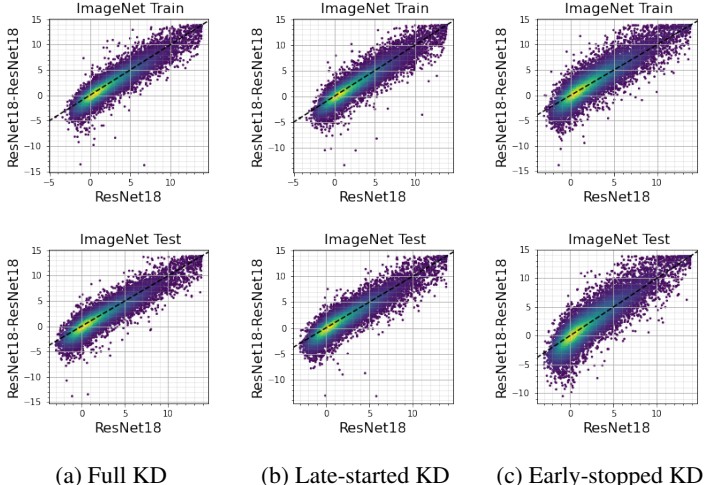

(a) Full KD        (b) Late-started KD      (c) Early-stopped KD

Figure 7: **Teacher-student logit plots for Imagenet experiments:** We conduct Imagenet self-distillation on ResNet18 in three different settings, involving full knowledge distillation, late-started distillation (from exactly mid-way through one-hot training) and early-stopped distillation (again, at the midway point, after which we complete with one-hot training). The plots for the training data are on top, and for test data in the bottom). Note that [7] recommend early-stopped distillation. We find underfitting of low-confidence points in all the settings, with the most underfitting in the last setting.

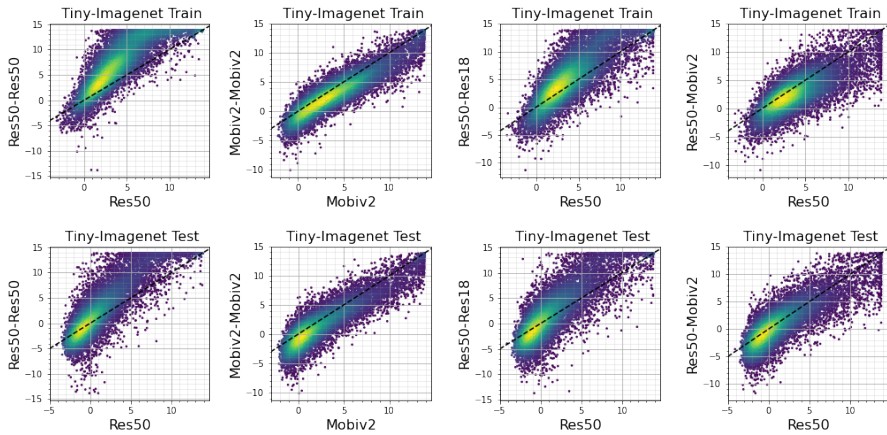

Figure 8: **Teacher-student logit plots for Tiny-Imagenet experiments:** We report plots for various distillation settings involving ResNet50, ResNet18 and MobileNet-v2 (training data on top, test data in the bottom). We find underfitting of the low-confidence points in all the settings. We also find overfitting of the high-confidence points when the student is a ResNet.

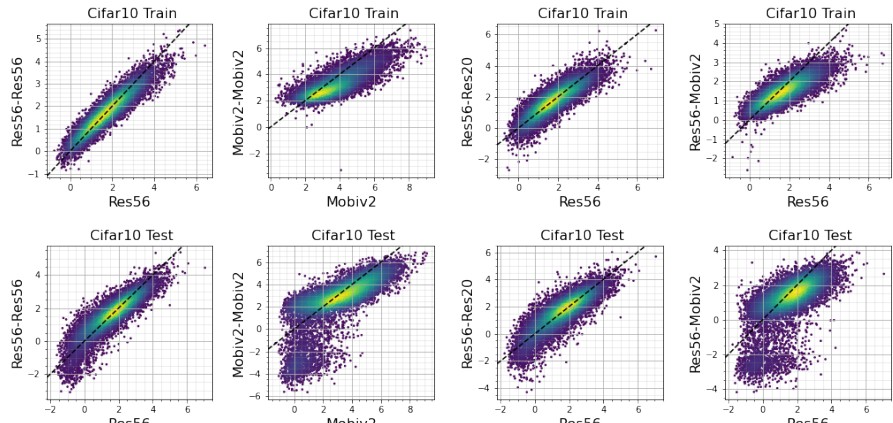

Figure 9: **Teacher-student logit plots for CIFAR10 experiments:** We report plots for various distillation settings involving ResNet56, ResNet20 and MobileNet-v2. We find that the underfitting phenomenon is almost non-existent in the training set (except for ResNet50 to ResNet20 distillation). However the phenomenon is prominent in the test dataset.

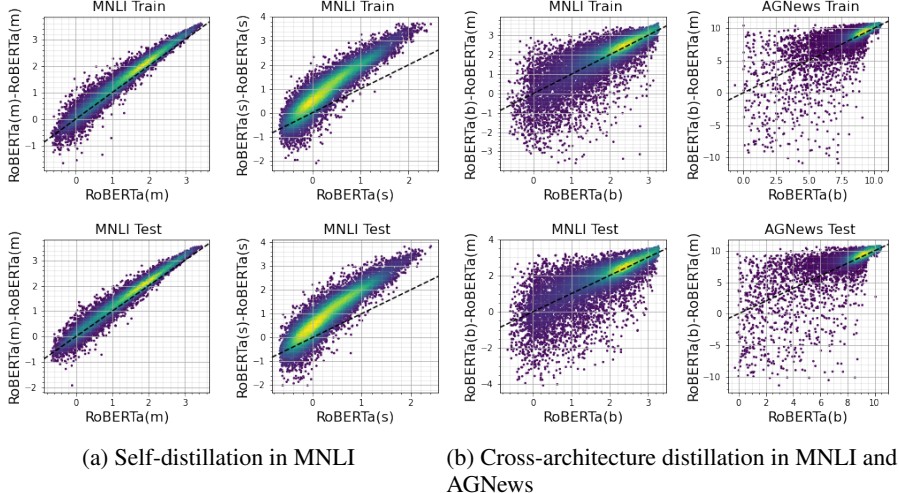

(a) Self-distillation in MNLI      (b) Cross-architecture distillation in MNLI and AGNews

Figure 10: **Teacher-student logit plots for MNLI and AGNews experiments:** We report plots for various distillation settings involving RoBERTa models. On the **left**, in the self-distillation settings on MNLI, we find significant underfitting of low-confidence points (and also overfitting), while high-confidence points are significantly overfit. On the **right**, we report cross-architecture (Base to Medium) distillation for MNLI and AGNews. Here, to a lesser extent, we see the same pattern. We interpret this as distillation reducing its "precision" on the lower-confidence points (perhaps by ignoring lower eigenvectors that provide finer precision).

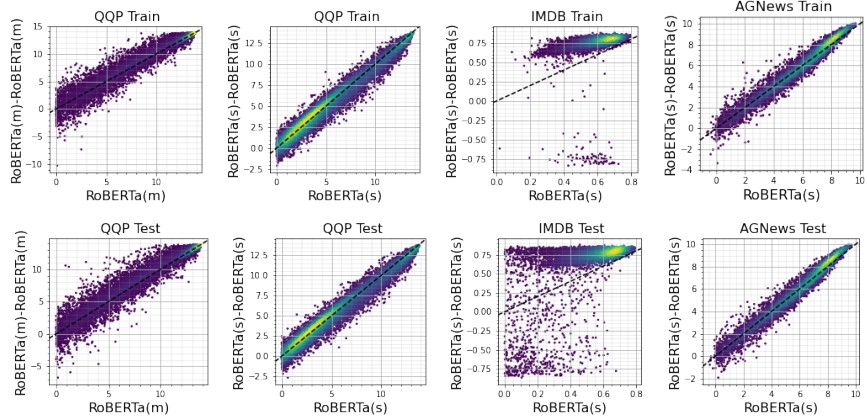

Figure 11: **Teacher-student logit plots for self-distillation in language datasets (QQP, IMDB, AGNews):** We report plots for various self-distillation settings involving RoBERTa models. Except for IMDB training dataset, we find both significant underfitting and overfitting for lower-confidence points (indicating lack of precision), and more precision for high-confidence points. For IMDB test and AGNews, there is an overfitting of the high-confidence points.

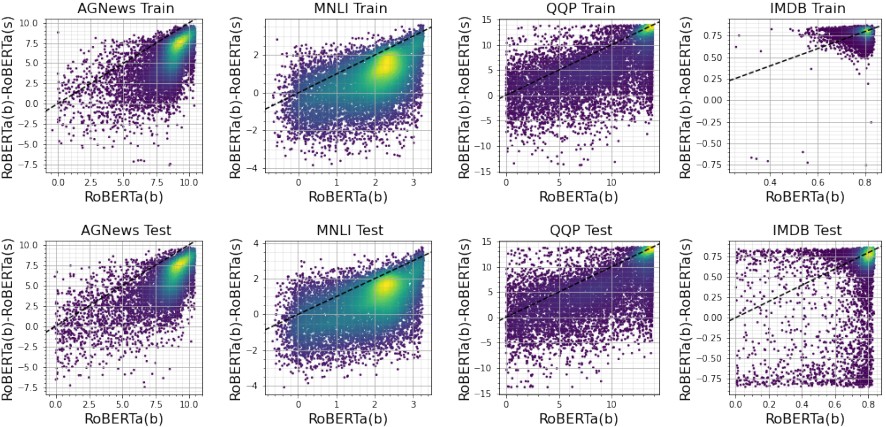

Figure 12: **Teacher-student logit plots for *cross-architecture* distillation in language datasets (AGNews, QQP, IMDB, MNLI):** We report plots for various cross-architecture distillation settings involving RoBERTa models. While we find significant student-teacher deviations in these settings, our typical patterns do not apply here. We believe that effects due to "cross-architecture gaps" may have likely drowned out the underfitting patterns, which is a more subtle phenomenon that shines in self-distillation settings.

### C.3 Teacher's predicted class vs. ground truth class

Recall that in all our scatter plots we have looked at the probabilities of the teacher and the student on the teacher's predicted class i.e., $(p_{y^{te}}^{te}(x), p_{y^{te}}^{st}(x))$ where $y^{te} \doteq \operatorname{argmax}_{y' \in [K]} p_{y'}^{te}(x)$. Another natural alternative would have been to look at the probabilities for the *ground truth class*, $(p_{y^\star}^{te}(x), p_{y^\star}^{st}(x))$ where $y^\star$ is the ground truth label. We chose to look at $y^{te}$ however, because we are interested in the "shortcomings" of the distillation procedure where the student only has access to teacher probabilities and not ground truth labels.

Nevertheless, one may still be curious as to what the probabilities for the ground truth class look like. First, we note that the plots look almost identical for the training dataset owing to the fact that the teacher model typically fits the data to low training error (we skip these plots to avoid redundancy). However, we find stark differences in the test dataset as shown in Fig 13. In particular, we see that the underfitting phenomenon is no longer prominent, and almost non-existent in many of our settings. This is surprising as this suggests that the student somehow matches the probabilities on the ground truth class of the teacher *despite not knowing what the ground truth class is*.

We note that previous work [33] has examined deviations on ground truth class probabilities albeit in an aggregated sense (at a class-level rather than at a sample-level). While they find that the student tends to have lower ground truth probability than the teacher on problems with label imbalance, they do *not* find any such difference on standard datasets without imbalance. This is in alignment with what we find above.

To further understand the underfit points from Sec C.2 (where we plot the probabilities on teacher's predicted class), in Fig 14, we dissect these plots into four groups: these groups depend on which amongst the teacher and student model classify the point correctly (according to ground truth). We consistently find that the underfit set of points is roughly *the union* of the set of all points where *at least one of the models is incorrect*. This has two noteworthy implications. First, in its attempt to deviate from the teacher, the student *corrects* some of the teacher's mistakes. But in doing so, the student also introduces *new mistakes* the teacher originally did not make. We conjecture that these may correspond to points which are inherently fuzzy e.g., they are similar to multiple classes.

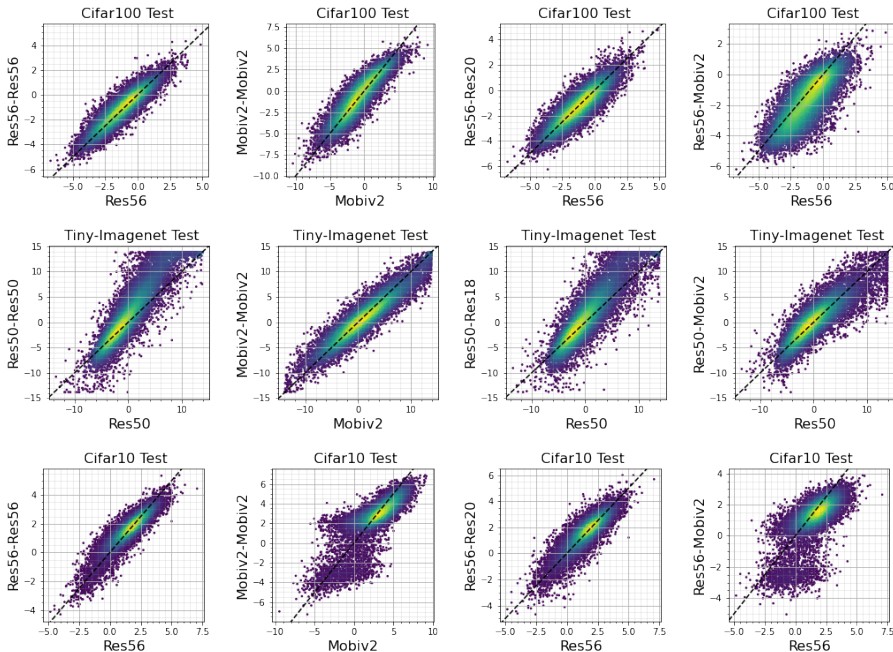

Figure 13: **Scatter plots for ground truth class:** Unlike in other plots where we report the probabilities for the class predicted by the teacher, here we focus on the ground truth class. Recall that the $X$-axis corresponds to the teacher, the $Y$-axis to the student, and all the probabilities are log-transformed. Surprisingly, we observe a much more subdued underfitting here, with the phenomenon completely disappearing e.g., in CIFAR100 and CIFAR10 ResNet distillation. This suggests that the student preserves the ground-truth probabilities despite no knowledge of what the ground-truth class is, while underfitting on the teacher's predicted class.

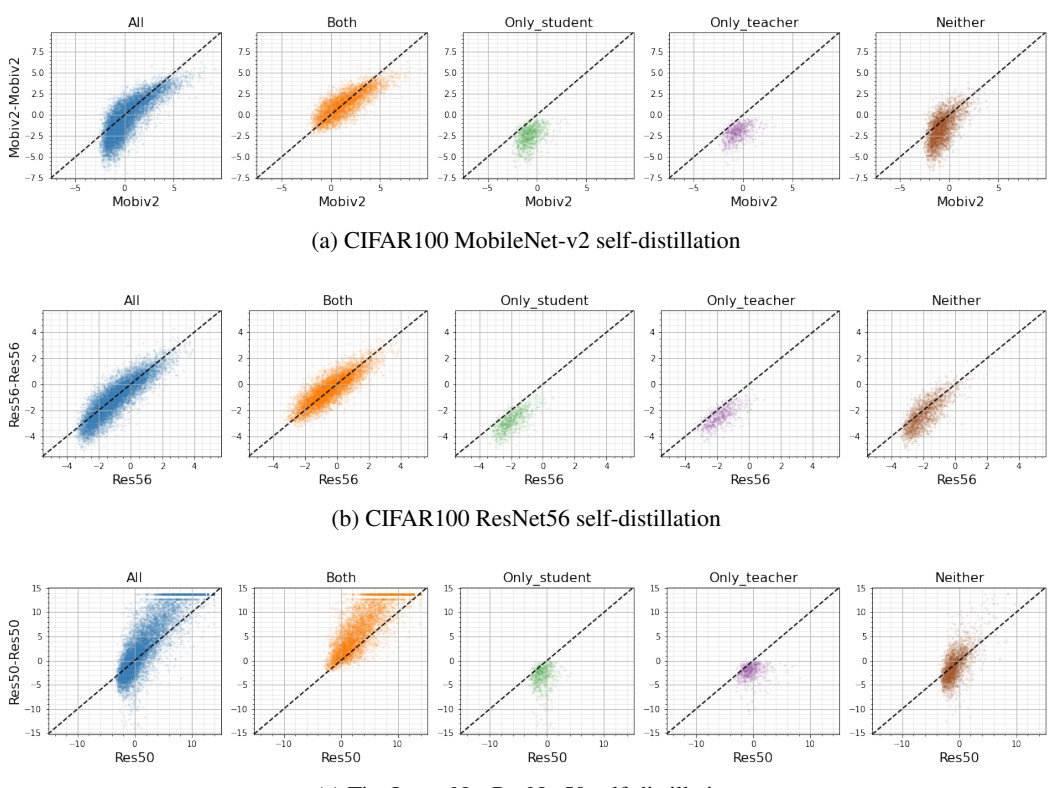

(a) CIFAR100 MobileNet-v2 self-distillation

(b) CIFAR100 ResNet56 self-distillation

(c) TinyImageNet ResNet50 self-distillation

Figure 14: **Dissecting the underfit points:** Across a few settings on TinyImagenet and CIFAR100, we separate the teacher-student scatter plots of logit-transformed probabilities (for teacher's top predicted class) into four subsets: subsets where both models' top prediction is correct (titled as "Both"), where only the student gets correct ("Only_student"), where only the teacher gets correct ("Only_teacher"), where neither get correct ("Neither"). We consistently find that the student's underfit points are points where at least one of the models go wrong.

Table 3: **Quantification of confidence exaggeration for *self*-distillation settings on *image* datasets:** Slope greater than 1 implies confidence exaggeration. Slope is computed for bottom 25% by teacher's confidence.

| Dataset | Teacher | Student | Slope | |
|---|---|---|---|---|
| | | | **Train** | **Test** |
| CIFAR10 | MobileNet-v2-1.0 | MobileNet-v2-1.0 | 0.22 | 1.37 |
| | ResNet-56 | ResNet-56 | 0.87 | 1.13 |
| CIFAR100 | MobileNet-v2-1.0 | MobileNet-v2-1.0 | 0.80 | 1.22 |
| | ResNet-56 | ResNet-56 | 1.26 | 1.22 |
| (noisy) | ResNet-56 | ResNet-56 | 1.55 | 1.19 |
| (v2 hyperparameters) | ResNet-56 | ResNet-56 | 1.25 | 1.31 |
| Tiny-ImageNet | MobileNet-v2-1.0 | MobileNet-v2-1.0 | 1.24 | 1.22 |
| | ResNet-50 | ResNet-50 | 1.97 | 1.20 |
| ImageNet | ResNet-18 | ResNet-18 (full KD) | 1.27 | 1.22 |
| | ResNet-18 | ResNet-18 (late KD) | 1.26 | 1.24 |
| | ResNet-18 | ResNet-18 (early KD) | 1.38 | 1.37 |

Table 4: **Quantification of confidence exaggeration for *cross*-distillation settings on *image* datasets:** Slope greater than 1 implies confidence exaggeration. Slope is computed for bottom 25% by teacher's confidence.

| Dataset | Teacher | Student | Slope | |
|---|---|---|---|---|
| | | | **Train** | **Test** |
| CIFAR10 | ResNet-56 | MobileNet-v2-1.0 | 0.57 | 1.18 |
| | ResNet-56 | ResNet-20 | 1.05 | 1.16 |
| CIFAR100 | ResNet-56 | MobileNet-v2-1.0 | 0.95 | 1.03 |
| | ResNet-56 | ResNet-20 | 1.26 | 1.12 |
| (noisy) | ResNet-56 | ResNet-20 | 1.50 | 1.60 |
| Tiny-ImageNet | ResNet-50 | MobileNet-v2-1.0 | 1.29 | 1.08 |
| | ResNet-50 | ResNet-18 | 1.69 | 1.23 |

### C.4 Quantification of exaggeration

Although we report the exaggeration of confidence levels as a qualitative observation, we attempt a quantification for the sake of completeness. To this end, our idea is to fit a least-squares line $Y = mX + c$ through the scatter plots of $(\phi(p^{\mathsf{te}}_{y^{\mathsf{te}}}(x)), \phi(p^{\mathsf{st}}_{y^{\mathsf{te}}}(x)))$ and examine the slope of the line. If $m > 1$, we infer that there is an exaggeration of confidence values. Note that this is only a proxy measure and may not always fully represent the qualitative phenomenon.

In the image datasets, recall that this phenomenon most robustly occurred in the teacher's low-confidence points. Hence, we report the values of the slope for the bottom 25%-ile points, sorted by the teacher's confidence $\phi(p^{\mathsf{te}}_{y^{\mathsf{te}}}(x))$. Table 3 corresponds to self-distillation and Table 4 to cross-architecture. These values faithfully capture our qualitative observations. In all the image datasets, on test data, the slope *is* greater than 1. The same holds on training data in a majority of our settings, except for the CIFAR-10 settings, and the CIFAR100 settings with a MobileNet student, where we did qualitatively observe the lack of confidence exaggeration.

For the language datasets, recall that there was both an underfitting and overfitting of low-confidence points, but an overfitting of the high-confidence points. We focus on the latter and report the values of the slope for the top 25%-ile points, Table 5 corresponds to self-distillation and Table 6 to cross-architecture. On test data, the slope is larger than 1 for seven out of our 12 settings. However, we note that we do not see a perfect agreement between these values and our observations from the plots e.g., in IMDB test data, self-distillation of RoBERTa-small, the phenomenon is strong, but this is not represented in the slope.

Table 5: **Quantification of confidence exaggeration for *self*-distillation settings on *language* datasets:** Slope greater than 1 implies confidence exaggeration. Slope is computed for top 25% points by teacher's confidence.

| Dataset | Teacher | Student | Slope | |
|---------|---------|---------|-------|------|
| | | | **Train** | **Test** |
| MNLI | RoBERTa-Small | RoBerta-Small | 1.28 | 1.30 |
| | RoBERTa-Medium | RoBerta-Medium | 0.98 | 1.00 |
| IMDB | RoBERTa-Small | RoBerta-Small | 0.37 | 0.38 |
| QQP | RoBERTa-Small | RoBerta-Small | 1.02 | 1.01 |
| | RoBERTa-Medium | RoBerta-Medium | 0.54 | 0.59 |
| AGNews | RoBERTa-Small | RoBerta-Small | 1.03 | 1.02 |

Table 6: **Quantification of confidence exaggeration for *cross*-distillation settings on *language* datasets:** Slope greater than 1 implies confidence exaggeration. Slope is computed for top 25% of points by teacher's confidence.

| Dataset | Teacher | Student | Slope | |
|---------|---------|---------|-------|------|
| | | | **Train** | **Test** |
| MNLI | RoBERTa-Base | RoBerta-Small | 1.69 | 1.68 |
| | RoBERTa-Base | RoBerta-Medium | 1.10 | 1.19 |
| IMDB | RoBERTa-Base | RoBerta-Small | −0.70 | 0.60 |
| QQP | RoBERTa-Base | RoBerta-Small | 23.20 | 21.53 |
| AGNews | RoBERTa-Base | RoBerta-Small | 0.90 | 1.10 |
| | RoBERTa-Base | RoBerta-Medium | 0.88 | 0.88 |

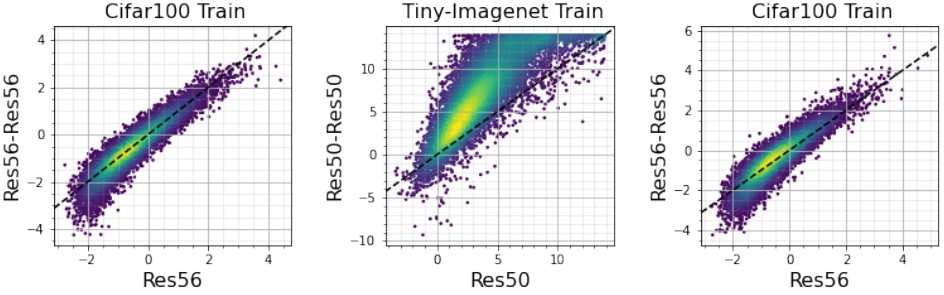

Figure 15: **Underfitting holds for longer runs and for smaller batch sizes:** For the self-distillation setting in CIFAR100 and TinyImagenet **(left two figures)**, we find that the student underfits teacher's low-confidence points even after an extended period of training (roughly $2\times$ longer). On the **right**, we find in the CIFAR100 setting that underfitting occurs even for smaller batch sizes.

## C.5  Ablations

We provide some additional ablations in the following section.

**Longer training:** In Fig 15 (left two images), we conduct experiments where we run knowledge distillation with the ResNet-56 student on CIFAR100 for $2.3\times$ longer ($50k$ steps instead of $21.6k$ steps overall) and with the ResNet-50 student on TinyImagenet for about $2\times$ longer ($300k$ steps over instead of roughly $150k$ steps). We find the resulting plots to continue to have the same underfitting as the earlier plots. It is worth noting that in contrast, in a linear setting, it is reasonable to expect the underfitting to disappear after sufficiently long training. Therefore, the persistent underfitting in the non-linear setting is remarkable and suggests one of two possibilities:

- The underfitting is persistent simply because the student is not trained sufficiently long enough i.e., perhaps, when trained $10\times$ longer, the network might end up fitting the teacher probabilities perfectly.

- The network has reached a local optimum of the knowledge distillation loss and can never fit the teacher precisely. This may suggest an added regularization effect in distillation, besides the eigenspace regularization.

**Smaller batch size/learning rate:** In Fig 15 (right image), we also verify that in the CIFAR100 setting if we set peak learning rate to $0.1$ (rather than $1.0$) and batch size to $128$ (rather than $1024$), our observations still hold. This is in addition to the second hyperparameter setting for CIFAR100 in Fig 6.

**A note on distillation weight.** For all of our students (except in ImageNet), we fix the distillation weight to be $1.0$ (and so there is no one-hot loss). This is because we are interested in studying deviations under the distillation loss; after all, it is most surprising when the student deviates from the teacher when trained on a pure distillation loss which disincentivizes any deviations.

Nevertheless, for ImageNet, we follow Cho and Hariharan [7] and set the distillation weight to be small, at $0.1$ (and correspondingly, the one-hot weight to be $0.9$). We still observe confidence exaggeration in this setting in Fig 7. Thus, the phenomenon is robust to this hyperparameter.

**Scatter plot for other metrics:** So far we have looked at student-teacher deviations via scatter plots of the probabilities on the teacher's top class, after applying a logit transformation. It is natural to ask what these plots would look like under other variations. We explore this in Fig 16 for the CIFAR100 ResNet-56 self-distillation setting.

For easy reference, in the top left of Fig 16, we first show the standard logit-transformed probabilities plot where we find the underfitting phenomenon. In the second top figure, we then directly plot the probabilities instead of applying the logit transformation on top of it. We find that the underfitting phenomenon does not prominently stand out here (although visible upon scrutiny, if we examine below the $X = Y$ line for $X \approx 0$). This illegibility is because small probability values tend to concentrate around 0; the logit transform however acts as a magnifying lens onto the behavior of

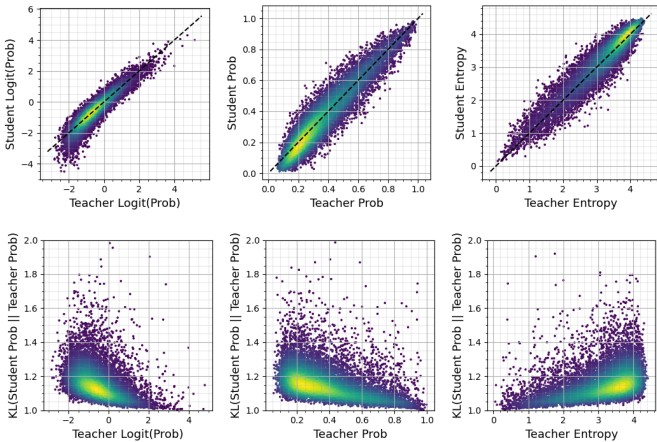

Figure 16: **Scatter plots for various metrics:** While in the main paper we presented scatter plots of logit-transformed probabilities, here we present scatter plots for various metrics, including the probabilities themselves, entropy of the probabilities, and the KL divergence of the student probabilities from the teacher. We find that the KL-divergence plots capture similar intuition as our logit-transformed probability plots. On the other hand, directly plotting the probabilities themselves is not as visually informative.

small probability values. For the third top figure, we provide a scatter plot of entropy values of the teacher and student probability values to determine if the student distinctively deviates in terms of entropy from the teacher. It is not clear what characteristic behavior appears in this plot.

In the bottom plots, on the $Y$ axis we plot the KL-divergence of the student's probability from the teacher's probability. Along the $X$ axis we plot the same quantities as in the top row's three plots. Here, across the board, we observe behavior that is aligned with our earlier findings: the KL-divergence of the student tends to be higher on teacher's lower-confidence points, where "lower confidence" can be interpreted as either points where its top probability is low, or points where the teacher is "confused" enough to have high entropy.

Table 7: Summary of the more general training settings used to verify our theoretical claim.

| Hyperparameter | Noisy-MNIST/RandomFeatures | MNIST/MLP | CIFAR10/CNN |
|---|---|---|---|
| Width | 5000 ReLU Random Features | 1000 | 100 |
| Kernel | - | - | $(6, 6)$ |
| Max pool | - | - | $(2, 2)$ |
| Depth | 1 | 2 | 3 |
| Number of Classes | 10 | 10 | 10 |
| Training data size | 128 | 128 | 8192 |
| Batch size | 128 | 32 | 128 |
| Epochs | 40 | 20 | 40 |
| Label Noise | 25% (uniform) | None | None |
| Learning rate | $10^{-3}$ | $10^{-4}$ | $10^{-4}$ |
| Distillation weight | 1.0 | 1.0 | 1.0 |
| Distillation temperature | 4.0 | 4.0 | 4.0 |
| Optimizer | Adam | Adam | Adam |

## D    Further experiments verifying eigenspace regularization

### D.1    Description of settings

In this section, we demonstrate the theoretical claims in §4 in practice even in situations where our theoretical assumptions do not hold good. We go beyond our assumptions in the following ways:

1. We consider three architectures: a linear random features model, an MLP and a CNN.
2. All are trained with the cross-entropy loss (instead of the squared error loss).
3. We consider multi-class problems instead of scalar-valued problems.
4. We use a finite learning rate with minibatches and Adam.
5. We test on a noisy-MNIST dataset, MNIST and CIFAR10 dataset.

We provide exact details of these three settings in Table 7.

### D.2    Observations

Through the following observations in our setups above, we establish how our insights generalize well beyond our particular theoretical setting:

1. In all these settings, the student fails to match the teacher's probabilities adequately, as seen in Fig 18. This is despite the fact that they both share the same representational capacity. Furthermore, we find a systematic underfitting of the low-confidence points.
2. At the same time, we also observe in Fig 19, Fig 20, Fig 21 that the convergence rate of the student is much faster along the top eigendirections when compared to the teacher in nearly all the pairs of eigendirections that we randomly picked to examine. See §D.3 for how exactly these plots are computed. Note that these plots are shown for the first layer parameters (with respect to the eigenspace of the raw inputs). We show some preliminary evidence that these can be extended to subsequent layers as well (see Fig 22, 23).
3. We also confirm the claim we made in Sec 5.1 to connect the exaggeration of confidence levels to the exaggeration of bias in the eigenspace. In Fig 18 (left), we see that on the mislabeled examples in the NoisyMNIST setting, the teacher has low confidence; the student has even lower confidence on these points. For the sake of completeness, we also show that these noisy examples are indeed fit by the bottom eigendirections in Fig 17. Thus, naturally, a slower convergence along the bottom eigendirections would lead to underfitting of the mislabeled data.

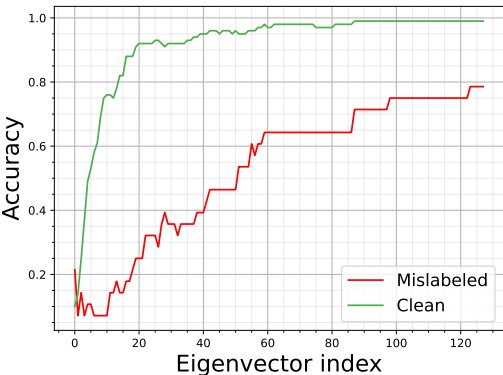

Figure 17: **Bottom eigenvectors help fit mislabeled data:** For the sake of completeness, in the NoisyMNIST setting we report how the accuracy of the model ($Y$ axis) degrades as we retain only components of the weights along the top $K$ eigendirections ($K$ corresponds to $X$ axis). The accuracy on the mislabeled data, as expected, degrades quickly as we lose the bottommost eigenvectors, while the accuracy on clean data is preserved even until $K$ goes as small as 20.

Thus, our insights from the linear regression setting in §4 apply to a wider range of settings. We also find that underfitting happens in these settings, reinforcing the connection between the eigenspace regularization effect and underfitting.

### D.3    How eigenvalue trajectories are plotted

**How eigendirection trajectories are constructed.**

In our theoretical analysis, we looked at how the component of the weight vector along a data eigendirection would evolve over time. To study this quantity in more general settings, there are two generalizations we must tackle. First, we have to deal with weight matrices or tensors rather than vectors. Next, for the second and higher-layer weight matrices, it is not clear what corresponding eigenspace we must consider, since the corresponding input is not fixed over time.

Below, we describe how we address these challenges. Our main results in Fig 19, Fig 20, Fig 21 are focused on the first layer weights, where the second challenge is automatically resolved (the eigenspace is fixed to be that of the fixed input data). Later, we show some preliminary extensions to subsequent layers.

**How data eigendirections are computed.** For the case of the linear model and MLP model, we compute the eigendirections $\mathbf{v}_1, \mathbf{v}_2, \ldots \in \mathbb{R}^d$ directly from the training input features. Here, $p$ is the dimensionality of the (vectorized) data. In the linear model this equals the number of random features, and in the MLP model this is the dimensionality of the raw data (e.g., 784 for MNIST). For the convolutional model, we first take *random* patches of the images of the same shape as the kernel (say $(K, K, C)$ where $C$ is the number of channels). We vectorize these patches into $\mathbb{R}^p$ where $p = K \cdot K \cdot C$ before computing the eigendirections of the data.

**How weight components along eigendirections are computed.** First we transform our weights into a matrix $\mathbf{W} \in \mathbb{R}^{p \times h}$. For the linear and MLP model, we let $\mathbf{W} \in \mathbb{R}^{p \times h}$ be the weight matrix applied on the $p$-dimensional data. Here $h$ is the number of outputs of this matrix. In the case of random features, $h$ equals the number of classes, and in the case of the MLP, $h$ is the number of output hidden units of that layer. For the CNN, we flatten the 4-dimensional convolutional weights into $\mathbf{W} \in \mathbb{R}^{p \times h}$ where $p = K \cdot K \cdot C$. Here, $h$ is the number of output hidden units of that layer.

Having appropriately transformed our weights into a matrix $\mathbf{W}$, for any index $k$, we calculate the component of the weights along that eigendirection as $\mathbf{W}^T \mathbf{v}_k$; we further scalarize this as $\|\mathbf{W}^T \mathbf{v}_k\|_2$. For the plots, we pick two random eigendirections and plot the projection of the weights along those over the course of time.

**How to read the plots.** In all the plots, the bottom eigendirection is along the $Y$ axis, the top along the $X$ axis. The final weights of either model are indicated by a $\star$. When we say the model shows

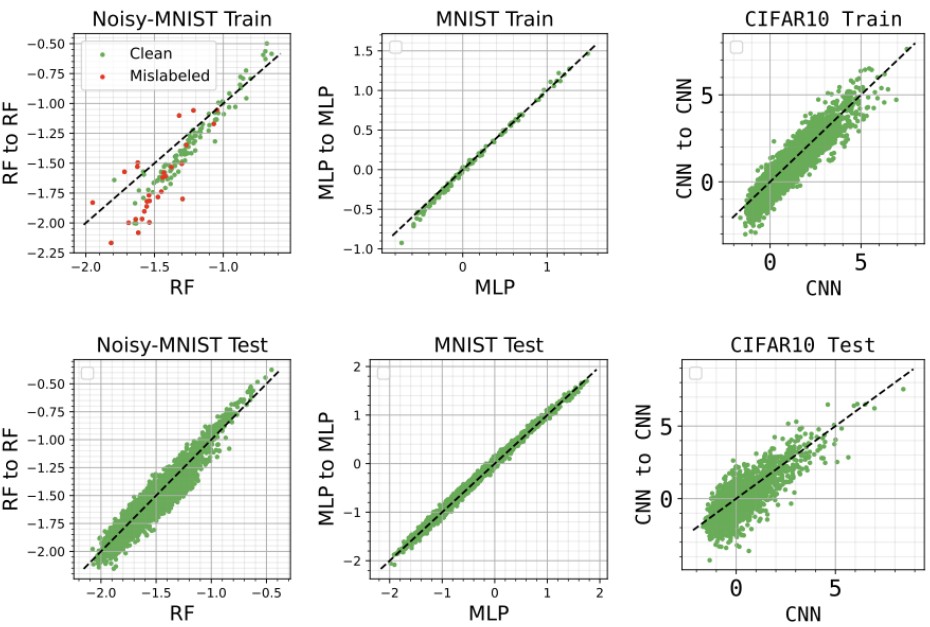

Figure 18: **Confidence exaggeration verifying our theory:** We plot the logit-logit scatter plots, similar to §3, for the three settings in §D — these are also the settings where we verify that distillation exaggerates the implicit bias. Each column corresponds to a different setting, while the top and bottom row correspond to train and test data respectively. Across all the three settings, we find low-confidence underfitting, particularly in the training dataset.

"implicit bias", we mean that it converges faster along the top direction in the $X$ axis than the $Y$ axis. This can be inferred by comparing what *fraction* of the $X$ and $Y$ axes have been covered at any point. Typically, we find that the progress along $X$ axis dominates that along the $Y$ axis. Intuitively, when this bias is extreme, the trajectory would reach its final $X$ axis value first with no displacement along the $Y$ axis, and only then take a sharp right-angle turn to progress along the $Y$ axis. In practice, we see a softer form of this bias where, informally put, the trajectory takes a "convex" shape. For the student however, since this bias is strong, the trajectory tends more towards the sharper turn (and is more "strongly convex").

**Extending to subsequent layers.** The main challenge in extending these plots to a subsequent layer is the fact that these layers act on a time-evolving eigenspace, one that corresponds to the hidden representation of the first layer at any given time. As a preliminary experiment, we fix this eigenspace to be that of the *teacher*'s hidden representation at the *end* of its training. We then train the student with the *same initialization* as that of the teacher so that there is a meaningful mapping between the representation of the two (at least in simple settings, all models originating from the same initialization are known to share interchangeable representations.) Note that we enforce the same initialization in all our previous plots as well. Finally, we plot the student and the teacher's weights projected along the fixed eigenspace of the teacher's representation.

### D.4 Verifying eigenspace regularization for random features on NoisyMNIST

Please refer Fig 19.

### D.5 Verifying eigenspace regularization for MLP on MNIST

Please refer Fig 20.

### D.6 Verifying eigenspace regularization for CNN on CIFAR10

Please refer Fig 21.

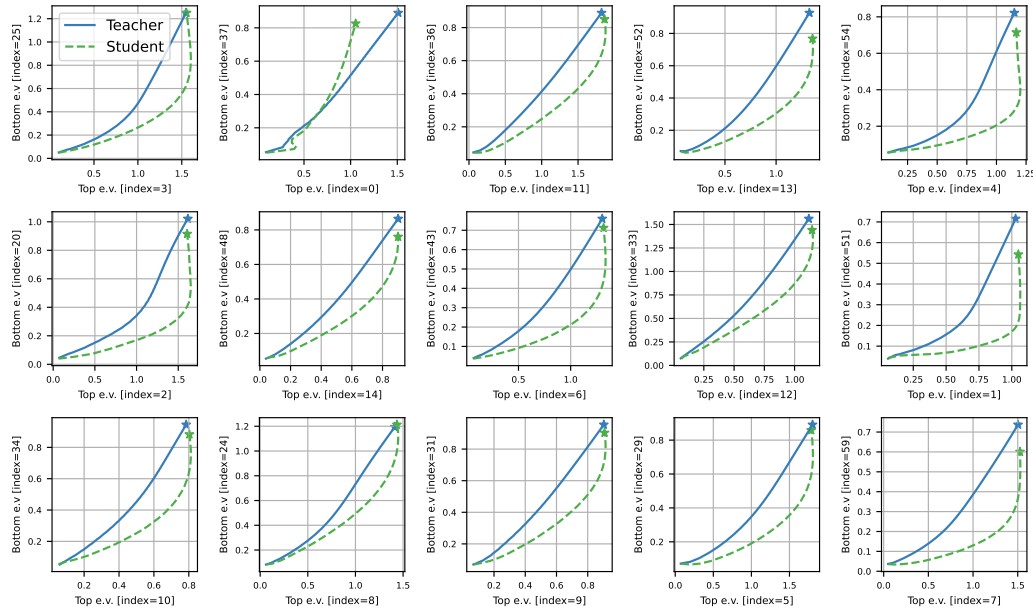

Figure 19: **Eigenspace convergence plots verifying the eigenspace theory for NoisyMNIST-RandomFeatures setting**: In all these plots, the $X$ axis corresponds to the top eigenvector and the $Y$ axis to the bottom eigenvector (see §D for how they are randomly picked). Each plot shows the trajectory projected onto the two eigendirections with the $\star$ corresponding to the final parameters. In all but one case we find that both the student and the teacher converge faster to their final $X$ value, than to their $Y$ value showing that both have a bias towards higher eigendirections. But importantly, this bias is exaggerated for the student in all cases (except the one case in top row, second column), proving our main theoretical claim in §4 in a more general setting with multi-class cross-entropy loss, finite learning rate etc., See §D for discussion.

## D.7 Extending to intermediate layers

Please refer Fig 22 and Fig 23.

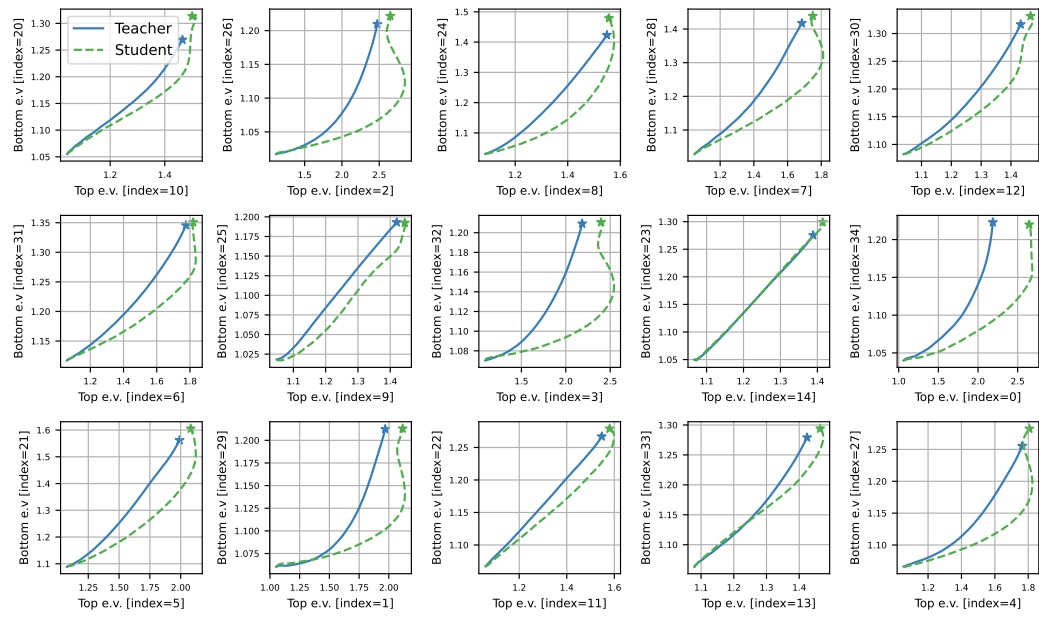

Figure 20: **Eigenspace convergence plots verifying the eigenspace theory for MNIST-MLP setting** : In all cases (except one), we find that the student converges faster to the final $X$ value of the teacher than it does along the $Y$ axis; in the one exceptional case (row 2, col 4), we do not see any difference. This demonstrates our main theoretical claim in §4 in a neural network setting. See §D for discussion.

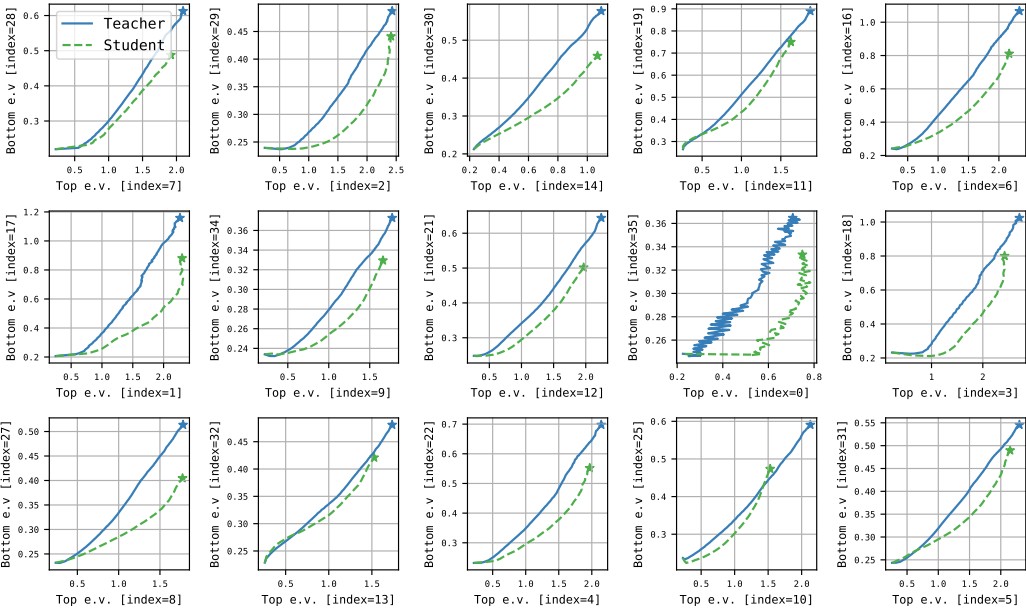

Figure 21: **Eigenspace convergence plots verifying the eigenspace theory for CIFAR10-CNN setting**: In *all* cases, we find that the student converges faster to the final $X$ value of the teacher than it does along the $Y$ axis. This demonstrates our main theoretical claim in §4 in a *convolutional* neural network setting. See §D for discussion.

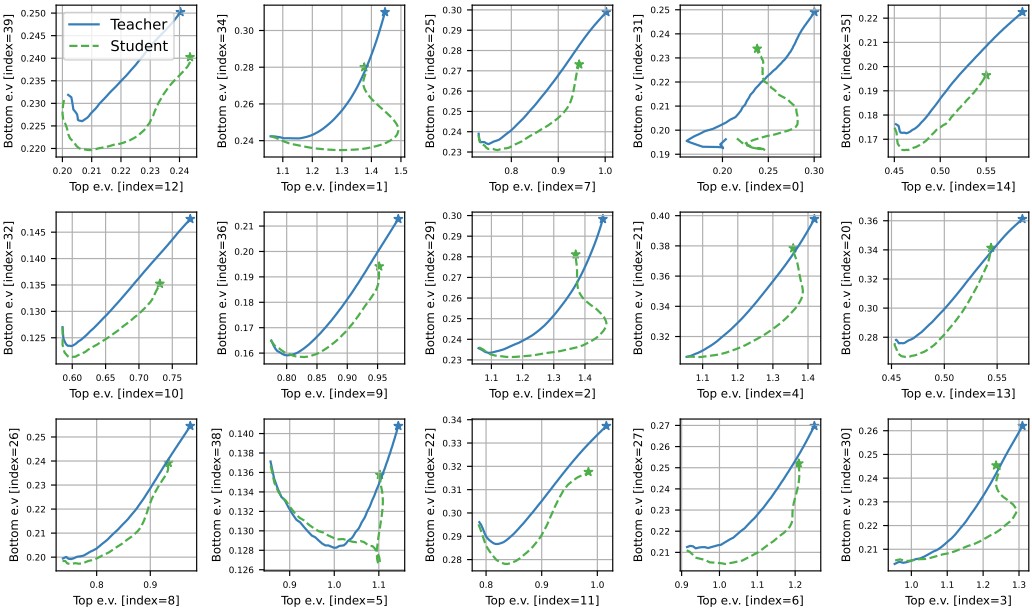

Figure 22: **Eigenspace convergence plots providing preliminary verification the eigenspace theory for the *intermediate* layer in the MNIST-MLP setting**: In all cases (except top row, fourth), we find that the student converges faster to the final $X$ value of the teacher than it does along the $Y$ axis. This demonstrates our main theoretical claim in §4 in an *hidden layer* of a neural network. Note that these plots are, as one would expect, less well-behaved than the first-layer plots in Fig 20. See §D for discussion.

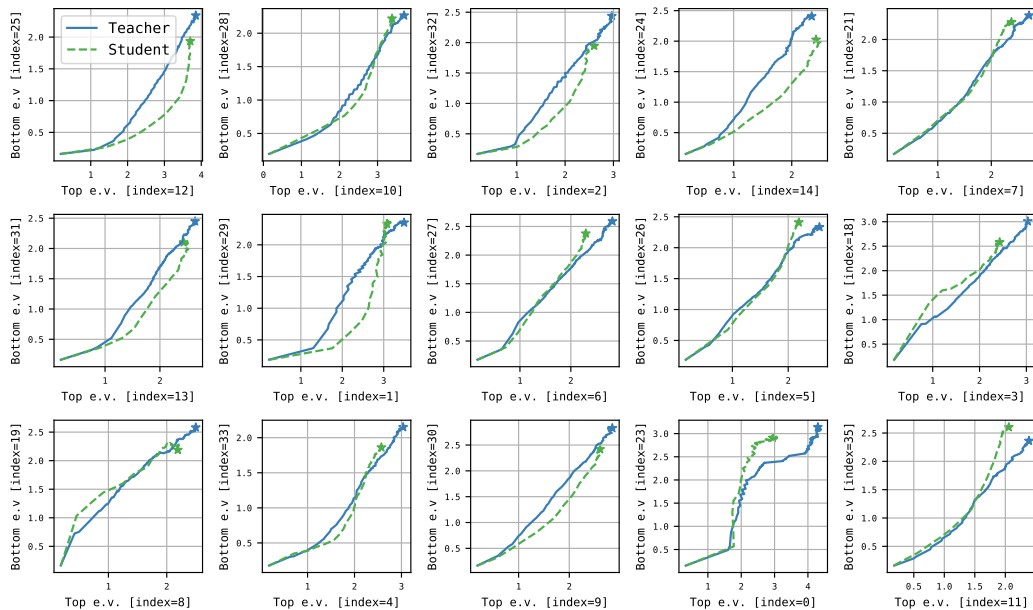

Figure 23: **Eigenspace convergence plots providing preliminary verification of the eigenspace theory for the *intermediate* layer CIFAR-CNN setting**: Here, we find that in a majority of the slices (indexed as 1,2,3,4,6,7,12 and 13 in row-major order), the student has an exaggerated bias than the teacher; in 5 slices (indexed as 2,5,8,9 and 12), there is little change in bias; in 4 slices the student shows a de-exaggerated bias than the teacher. Note that these plots are, as one would expect, less well-behaved than the first-layer plots in Fig 21. See §D for discussion.

# E Further experiments on loss-switching

In the main paper, we presented results on loss-switching between one-hot and distillation, inspired by prior work [7, 58, 21] that has proposed switching *from* distillation *to* one-hot. We specifically demonstrated the effect of this switch and the reverse, in a controlled CIFAR100 experiment, one with an interpolating and another with a non-interpolating teacher. Here, we present two more results: one with an interpolating CIFAR100 teacher in different hyperparameter settings (see v1 setting in §C.1) and another with a non-interpolating TinyImagenet teacher. These plots are shown in Fig 24. We also present how the logit-logit plots of the student and teacher evolve over time for both settings in Fig 25 and Fig 26.

We make the following observations for the CIFAR100 setting:

1. Corroborating our effect of the interpolating teacher in CIFAR100, we again find that even for this interpolating teacher, switching to one-hot in the middle of training surprisingly hurts accuracy.

2. Remarkably, we find that for CIFAR100 switching to distillation towards the end of training, is able to regain nearly all of distillation's gains.

3. Fig 26 shows that switching to distillation is able to introduce the confidence exaggeration behavior even from the middle of training; switching to one-hot is able to suppress this behavior.

Note that here training is supposed to end at $21k$ steps, but we have extended it until $30k$ steps to look for any long-term effects of the switch.

In the case of TinyImagenet,

1. For a distilled model, switching to one-hot in the middle of training increases accuracy beyond even the purely distilled model. This is in line with our hypothesis that such a switch would be beneficial under a non-interpolating teacher.

2. Interestingly, for a one-hot-trained model, switching to distillation *is* helpful enough to regain a significant fraction of distillation's gains. However, it does not gain as much accuracy as the distillation-to-one-hot switch.

3. Both the one-hot-trained model and the model which switched to one-hot, suffer in accuracy when trained for a long time. This suggests that any switch to one-hot must be done only for a short amount of time.

4. Fig 25 shows that switching to distillation is able to introduce the confidence exaggeration behavior; switching to one-hot is able to suppress this deviation. This replicates the same observation we make for CIFAR-100 in Fig 26.

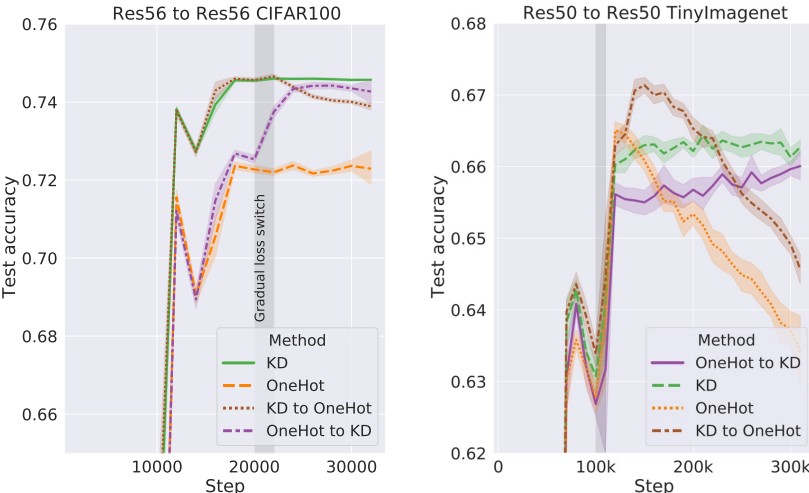

Figure 24: **Trajectory of test accuracy for loss-switching over longer periods of time:** We gradually change the loss for our self-distillation settings in CIFAR100 and TinyImagenet and extend training for a longer period of time. Note that the teacher for the CIFAR100 setting is interpolating while that for the TinyImagenet setting is not. This results in different effects when the student switchs to a one-hot loss, wherein it helps under the non-interpolating teacher and hurts for the interpolating teacher.

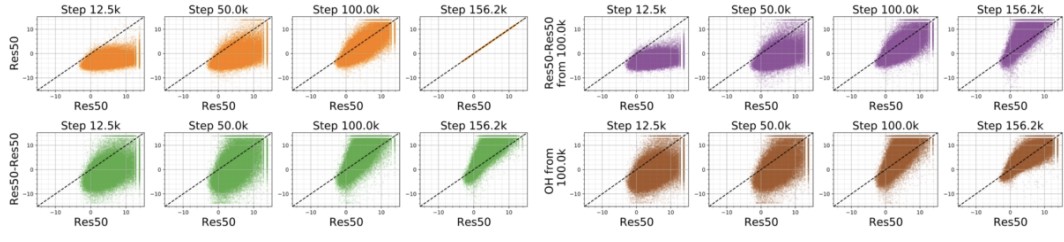

(a) One-hot and self-distillation.  (b) Loss-switching to distillation/one-hot at $100k$ steps.

Figure 25: **Evolution of student-teacher deviations over various steps of training for TinyImageNet ResNet50 self-distillation setup:** On the **left**, we present plots similar to §3 over the course of time for one-hot training (**top**) and distillation (**bottom**). On the **right**, we present similar plots with the loss switched to distillation (**top**) and one-hot (**bottom**) right after $100k$ steps. We observe that switching to distillation immediately introduces an exaggeration of the confidence levels.

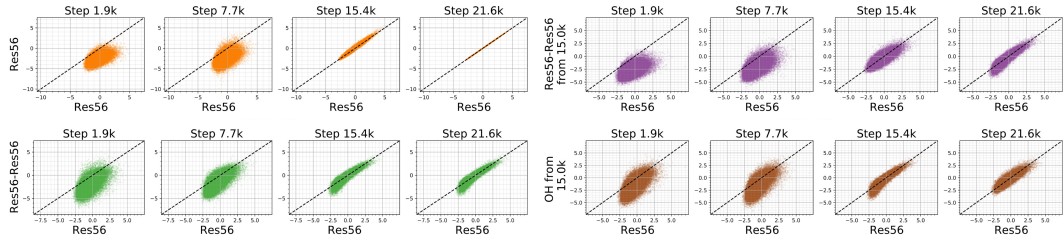

(a) One-hot and self-distillation.

(b) Loss-switching to distillation/one-hot at $15k$ steps.

Figure 26: **Evolution of logit-logit plots over various steps of training for CIFAR100 ResNet56 self-distillation setup:** On the **left**, we present plots for one-hot training (**top**) and distillation (**bottom**). On the **right**, we present similar plots the loss switched to distillation (**top**) and one-hot (**bottom**) at $15k$ steps. From the last two visualized plots in each, observe that switching to distillation introduces (a) underfitting of low-confidence points (b) while switching to one-hot curiously undoes this to an extent.

