# OpenReview forum: "On student-teacher deviations in distillation: does it pay to disobey?"
_NeurIPS.cc/2023/Conference — NeurIPS 2023 poster_

### Official Review · Reviewer_tbs9 · 2023-07-02

**Soundness:** 3 good
**Presentation:** 4 excellent
**Contribution:** 3 good
**Rating:** 7
**Confidence:** 4

**Summary:**

The paper performs a careful analysis of the discrepancy between the predictions made by the student and teacher in the context of knowledge distillation. Prior works has shown that (1) distillation improves student performance, so that it can sometimes outperform the teacher, and (2) the predictions made by the student are quite different from the predictions of the teacher in the end of distillation. The paper provides new insights into these observations, specifically:
- Student exaggerates the predicted probabilities of the teacher, i.e. it's less confident where the teacher is not confident and (sometimes) more confident where the teacher is confident
- The authors argue theoretically, and show empirical evidence that knowledge distillation exaggerates the inductive bias of gradient descent
- The authors show that knowledge distillation can hurt when the teacher is not interpolating the training data

**Strengths:**

**S1.** The authors perform careful high-quality analysis with targeted experiments highlighting specific conclusions. For example, the observation that the low-end of the confidence distribution for the student underfits the teacher (section 3) is demonstrated very thoroughly on a range of image and text problems, with nice figures, such as Fig. 2.

**S2.** The theory in Section 4 is presented very clearly and easy to follow.

**S3.** The experiment in Fig 3(a) is quite interesting, showing that the student is less confident than the teacher on almost all of the mislabeled datapoints.



**Weaknesses:**

**W1.** Some of the confidence scatter-plots in the appendix are less clean / interpretable than the ones in the main text. For example, Figure 12. Even in some panels in Fig. 5 the effect is not completely obvious.

**W2.** It would be very interesting to see how the training length impacts the observed confidence distribution. One of the main conclusions of [1] is that distillation performance keeps improving with longer training.

**W3.** [1] is actually reporting successful distillation results on ImageNet, while the authors argue in section 5.2 that distillation is known to hurt on ImageNet. What's causing the difference in the results?

**W4.** The theory is fairly simplistic, and only applies to linear models, and infinitesimal learning rates. However, the authors verify that some conclusions transfer to neural networks. But it is not very clear why they translate in the way they do. In particular, it's unclear why the authors measure the projection of the weights in the first layer on the eigenvalues of the covariance matrix of the data? Intuitively,  I would expect the analogy of the result for linear models would be some result on converging along the eigenvectors of the Hessian of the loss instead (although, it is not constant wrt weights, so it's not clear what that would mean exactly)?

**References**

[1] [_Knowledge Distillation: A Good Teacher Is Patient and Consistent_](https://openaccess.thecvf.com/content/CVPR2022/html/Beyer_Knowledge_Distillation_A_Good_Teacher_Is_Patient_and_Consistent_CVPR_2022_paper.html)

**Questions:**

**Q1.** Why do you think you see more student overcondfidence in language tasks compared to vision (line 140)?

**Limitations:**

Limitations addressed adequately.

---

> ### Author Rebuttal · Authors · 2023-08-04
>
>
> Thank you for appreciating the breadth of our experiments and the clarity in our theory. Also, thanks for the very detailed review!
>
> ----
> > Why do you think you see more student overconfidence in language tasks compared to vision (line 140)?
>
> Indeed, this is a curious phenomenon — thanks for paying close attention to our observations! We suspect this may have to do with there being only two to four classes in language tasks unlike the more complex image tasks that have 10s or 100s of classes. This would change the magnitude of the probabilities of various classes, and thus change the dynamics of learning qualitatively (while interestingly, still preserving the exaggeration phenomenon in some way or the other).
>
> ----
>
> > Some of the confidence scatter-plots in the appendix are less clean / interpretable than the ones in the main text. For example, Figure 12. Even in some panels in Fig. 5 the effect is not completely obvious.
>
> First of all, thanks again for examining all our results carefully!
>
> For Fig 5, we can quantitatively say that underfitting is prevalent here barring a couple of exceptions (e.g., Mobile-net self-distillation CIFAR100): In Table 3 and 4, we show that if we were to fit the bottom 25% of the teacher’s confident points, the slope is consistently > 1 in nearly all these plots (across test and train), except a few.
>
> For Fig 12, we completely agree with you — we even note this in the caption. **However, this is a cross-architecture setting, where capacity mismatch can induce confounding deviations that are not covered by the bias exaggeration theory!** We reported cross-architecture settings for the sake of completeness.
>
> We mention these exceptions in the main paper (lines 150 and 154) and discuss all of them clearly from line 661 in Appendix.
>
> ----
>
> > It would be very interesting to see how the training length impacts the observed confidence distribution. One of the main conclusions of [1] is that distillation performance keeps improving with longer training.
>
> This is a valuable ablation. **We have done this ablation in C.5, Lines 718-730.** We find that training longer still doesn’t fix the exaggeration. This is in line with Stanton et al., (“Does KD really work?”, Fig 6a) who find that significantly training longer only increases the student-teacher agreement on training data by a meager 2%! Possibly, the initial eigenspace regularization provided by KD traps the model in a local minimum of KD loss in the non-convex regime. This is an exciting fundamental question about distillation theory that our paper gives rise to.
>
> > [1] is actually reporting successful distillation results on ImageNet, while the authors argue in section 5.2 that distillation is known to hurt on ImageNet. What's causing the difference in the results?
>
> Our point was that **the _standard_ distillation recipe doesn’t work on ImageNet, as Cho and Hariharan note**. But there are certainly other sophisticated recipes that can get it working. For example, [1] uses augmentations, mixup and training for an extensively long time. We will make this clearer.
>
> ---
>
> > The authors verify that some conclusions transfer to neural networks. But it is not very clear why they translate in the way they do. In particular, it's unclear why the authors measure the projection of the weights in the first layer on the eigenvalues of the covariance matrix of the data?
>
> You’ve raised an interesting subtlety. There are multiple possible ways to informally generalize the theorem to the non-convex setting. In your interpretation, the theorem says: “there is an exaggerated bias when we project the weights onto the fixed Hessian of the loss landscape”.
>
> Our interpretation is: “there is an exaggerated bias when we project the weights onto the (fixed) eigenspace of the data”. **To us, this was the most natural interpretation as it follows from the manner of the proof.** To extend this to an MLP, we made some intuitive choices:
>
> 1. We consider each layer as an individual set of weights, and plot the eigenspace trajectories specific to those weights (e.g., D.5 & D.6 for first layer, and D.7 for intermediate layer)
> 2. We consider the “data” as the previous layer’s output. For the first layer, the data is the raw features themselves. For an intermediate layer, this would be the previous layer’s representation. But since we want this data to be fixed over time, we simply choose the representation computed at the end of teacher’s training. (Note that both the student and teacher start from the same initialization, so these representations become compatible.)
> 3. Since the weights are matrices, ideally, we would have to plot the eigentrajectory of each row of weights in this matrix. This is infeasible. So, we perform an l2 norm reduction so that we can visualize a single trajectory.
>
> Hopefully, this clarifies how we employ a reasonable informal generalization of the theorem. Note that in this framework, we still find a clear and neat difference in the two trajectories as predicted by the theorem.
>
> --------
>
> References
> [1] Knowledge Distillation: A Good Teacher Is Patient and Consistent
> [2]: Cho and Hariharan, On the Efficacy of Knowledge Distillation
>
> ----
>
>
> **We hope you find our answers to your questions satisfactory. If so, we sincerely hope you’re able to view the paper in a more positive light and re-evaluate its score! Thank you again for your time and detailed review.**

---

> > ### Comment · Reviewer_tbs9 · 2023-08-20
> > **Thank you for the rebuttal!**
> >
> > Dear authors, thank you so much for the detailed rebuttal and providing new intuitions and clarifications! I am fully satisfied with the response. Given that I already vote for accept, I will keep my score, as I think it best describes my evaluation of the paper.

---

### Official Review · Reviewer_zo9s · 2023-07-05

**Soundness:** 4 excellent
**Presentation:** 4 excellent
**Contribution:** 4 excellent
**Rating:** 7
**Confidence:** 4

**Summary:**

This paper explores the paradox in knowledge distillation where a &ldquo;student&rdquo; network deviates from the &ldquo;teacher&rdquo; network&rsquo;s probabilities but still outperforms the teacher. The authors found that the student network exaggerates the teacher&rsquo;s confidence levels across various architectures and data types. They also discovered that distillation amplifies the implicit bias of gradient descent, leading to faster convergence along top data eigendirections. This exaggerated bias is suggested as the reason for the student&rsquo;s improved performance. The study bridges theory and practice, offering insights for future work to enhance distillation benefits by inducing careful deviations.


**Strengths:**

- The paper's introduction is excellently crafted, presenting the problem, hypothesis, structure of the work, and results in a clear and engaging manner.
- The overall quality of the research paper is exceptional.
- The paper is highly coherent, presenting complex ideas in an understandable way.
- The contributions made in this paper are innovative and original, as far as I am aware.
- The findings significantly enhance our understanding of knowledge distillation behavior.
- The visual aids in the paper are straightforward and easy to comprehend.
- The experimental evaluation is comprehensive and thorough.
- The mathematical notation used in the paper is succinct, clear, and well-selected.
- The supplementary material provides a wealth of additional experimental evaluations.
- The paper is beautifully composed, making it a pleasure to read.


**Weaknesses:**

I generally dislike scatterplots when the density of points is as high as in the presented figures. The region of highest density is saturated and it becomes impossible to distinguish between densities in saturated regions. Consider an *actual* density plot with a heatmap or a contourplot as choice of visualization instead.

This is purely subjective, but I feel that the authors tends to overuse italics. While it can be useful to highlight key terms, its frequent use in every sentence or every other sentence diminishes its effectiveness.

I found only a single typo on line 300 &ldquo;similarities [17, 36], Several&rdquo; -> &ldquo;similarities [17, 36]. Several&rdquo;.

Figure 1. (a): Why is this the only figure with a rasterized instead of a vectorized graphic?


**Questions:**

None

**Limitations:**

The authors address the limitations of their work in Appendix A.

---

> ### Author Rebuttal · Authors · 2023-08-04
>
> We are pleased to hear that you have enjoyed reading the paper. Thanks in particular for examining our supplementary results and for raising many positive points about work.
>
> > Consider an actual density plot with a heatmap or a contourplot as choice of visualization instead.
>
> You’re right that it is better to use a density plot. Currently we use a scatter plot, although with a transparency factor which is somewhere in between a scatter plot and a density plot. **We’ve provided some sample plots in our global response PDF. Please let us know if you think they can be improved further.**
>
> > Figure 1. (a): Why is this the only figure with a rasterized instead of a vectorized graphic?
>
> Note that we rendered all our scatter plots as rasterized png files since it was faster to load. The vectorized pdf made it significantly harder to scroll over the pages, since they try to render 1000s of points.
>
> > tend to overuse italics
>
> Absolutely a fair point -– we will fix this.
>
> And thanks for spotting the typo!
>
> **Given your accurate and detailed summary of the paper, if you feel comfortable about increasing your confidence score, we would highly appreciate that! Thanks again for your positive review.**

---

> > ### Comment · Reviewer_zo9s · 2023-08-11
> >
> > > You’re right that it is better to use a density plot. Currently we use a scatter plot, although with a transparency factor which is somewhere in between a scatter plot and a density plot. We’ve provided some sample plots in our global response PDF. Please let us know if you think they can be improved further.
> >
> > IMO the density plot is much more informative -- well done.
> >
> > > Note that we rendered all our scatter plots as rasterized png files since it was faster to load. The vectorized pdf made it significantly harder to scroll over the pages, since they try to render 1000s of points.
> >
> > Yes, I recall having this issue when rendering vectorized scatterplots.
> >
> > Having read the other reviews and rebuttals, I want to thank the authors again for their contributions! I found the work to be highly interesting. I will stand with my rating and increase my confidence from 3 to 4.

---

> > > ### Author Response · Authors · 2023-08-11
> > > **Thank you for valuing our contributions!**
> > >
> > > Dear reviewer,
> > >
> > > **Thank you for going over all the reviews/rebuttals and updating your confidence. We are pleased to hear that you find the work highly interesting!** Once again, thanks for your valuable feedback.

---

### Official Review · Reviewer_pE3g · 2023-07-06

**Soundness:** 3 good
**Presentation:** 3 good
**Contribution:** 2 fair
**Rating:** 5
**Confidence:** 4

**Summary:**

This paper aims to understand the counter-intuitive phenomenon that the
student can sometimes outperform the teacher in terms of generalization
performance even when it deviates from the teacher's soft-labels during
training. Using a linear regression model, the authors provide a
theoretical result that explains this behavior. That is, through early
stopping, the student further exaggerates the top eigendirections and
suppresses the lower eigendirections than the teacher. Thus, the student
may deviate from the teacher's soft labels but attain a more favorable
implicit bias on the top eigendirections. The paper then relies on
numerical results to demonstrate that the similar behaviors occurs in
non-linear neural networks beyond regression.

**Strengths:**

1. The regime where student deviates from the teacher is less explored.
The paper is among the first to provide theoretical understand of the
improved student performance in this regime.

2. The paper provides both theoretical results and ample numerical
results for the phonomenon.

**Weaknesses:**

1. The theoretical result is for linear regression. It is unclear how
the analysis can be exended to classification problems.

2. Compared to the implicit bias in linear regression (with early
stopping), there are other ways that neural network training for
classification problems can produce an implicit bias. Thus, it is
difficult to tell whether the insight revealed by the theoretical result
is the most prominent factor.

3. The de-emphasis of lower eigen-directions could also be a double-edge
sword. That is, if the second and third eigne-directions correspond to ``useful''
parameters to learn, then the de-emphasis reported by the theoretical
result may also hurt the student.

**Questions:**

1. I wish to hear what the authors' thoughts are regarding the weakness
points above, i.e., (i) whether the theoretical insights can be
generalized to classification problems; (ii) the relevance of other
types of implicit bias; and (iii) the setting where de-emphazing lower
eigen-directions may hurt.

2. Can the authors also comment on the use of temperature in
student-teacher training? Using temperature also alters the teacher's
soft-label, and therefore it can be seen as another form of student not
trying to perfectly match the teacher. It has been demonstrated to
improve the student performance.

Post rebuttal phase:

The reviewer wishes to thank the authors for their response, which clarifies the "double-edge sword" effect shown in some of the experiments. However, I feel that overall the impact of the implicit-bias/top-eigenvalue effect is not conclusive: it sometimes helps, but sometimes doesn't. It is unclear how these explanations will guide student-teacher training. Thus, I think I will keep my review score.

**Limitations:**

Some discussion on the limitation in the conclusion section.

---

> ### Author Rebuttal · Authors · 2023-08-04
>
> Thank you for taking the time to provide your feedback on our paper. You’ve raised some interesting questions, some of which the paper addresses. We explain why below.
>
> ------
>
> > whether the theoretical insights can be generalized to classification problems;.
>
> We’d like to note that **we have provided empirical proof of the theorem’s claim for 3 different classification settings spanning 3 different models (linear, MLP, CNN)  in Fig 1b and Appendix D**. Here, our theoretical assumptions break in many different ways (we apply cross entropy + finite learning rate + multiclass + a different optimizer). Yet, we find a stark difference in the trajectory of the student and the teacher in the eigenspace — exactly as predicted by the theorem. We agree that it’s unclear how to extend the precise technicalities of our proof to these settings. But we hope you are convinced that these experiments consistently highlight the generality of our insights.
>
> -----
>
> > The de-emphasis of lower eigen-directions could also be a double-edge sword. That is, if the second and third eigne-directions correspond to ``useful'' parameters to learn, then the de-emphasis reported by the theoretical result may also hurt the student.
>
> Absolutely! Thanks for raising such an insightful point! **Sec 5.2 is meant to address this double-edgeness**, but we will rephrase it to make this clearer.
>
> Sec 5.2 says that, due to certain confounding factors, regularization may not always be a good thing i.e., it can be excessive, or as you say, a double-edged sword. You bring up one such confounding factor: usefulness of intermediate eigendirections. We highlighted another: the teacher’s top 1 train accuracy.
>
> In fact, we can phrase our factor in terms of yours. One way a teacher may have low training accuracy is via aggressive early-stopping; hence the teacher may have used only the top eigendirection and not relied as much on, say, a useful 2nd or 3d direction. When a student learns from this imperfect teacher, it may further de-emphasize these useful 2nd or 3d directions and thus suffer even more in accuracy. Regularization is thus counter-productive here. **We hope you see that this is not a weakness of our theory, but rather a universal fact about regularization that we have prominently noted in the paper.**
>
> ---
>
> > Can the authors also comment on the use of temperature in student-teacher training? …it can be seen as another form of student not trying to perfectly match the teacher.
>
>
> Agreed! To phrase this in terms of our theory, the top eigenvector can be thought of as an approximate classifier that assigns fuzzy class memberships. Thus, to fit de-fuzzified, spikier memberships, we need to rely on lower eigenvectors. Hence, a reasonable hypothesis is: “learning softer version of the teacher's memberships => more reliance on top eigenvectors => more imperfect fit of the teacher”.
>
> But again, this is a double-edged sword: if we try to learn overly-softened versions of the teacher's class memberships, we over-emphasize some “noisy” tail classes that the teacher picked up by happenstance. This may again demand the use of lower eigenvectors. Finding the sweetspot level of softness and relating it to our theory requires a rigorous analysis and is certainly a very interesting question for future work.
>
> **Having said all that, our work focused on the same-temperature case because this is the setting where it’s hardest to explain why there is any student-teacher deviation at all!**
>
> ----
>
> > There are other ways that neural network training for classification problems can produce an implicit bias. Thus, it is difficult to tell whether the insight revealed by the theoretical result is the most prominent factor.
> This is a thought-provoking question!
>
> First, we re-iterate that in our MLP and CNN settings, we do see a clear difference in the eigenspace trajectories of the model (Appendix D).
> But it is indeed right to wonder if distillation also exaggerates other sorts of biases in neural networks. We believe our work has empirically/theoretically identified the first exaggeration of this type in GD-trained models. It seems reasonable to say that finding other such exaggerations is a great open question for future work to tackle based on our insights.
>
> —-----
>
> **We hope that our answers convince you as to how our paper addresses some of your questions already; and for the other questions hopefully our response gives you a satisfactory new insight.** We are eager to know if you’ll be able to re-evaluate our paper's score (and contribution score) in light of this discussion. Thanks again for the insightful questions! Please let us know if you have any further ones.

---

> > ### Comment · Reviewer_pE3g · 2023-08-11
> >
> > I thank the authors for their response. The discussion in Section 5.2 does read more like "a double-edge sword" now, and thanks for pointing out this interpretation. However, I feel that overall the impact of the implicit-bias/top-eigenvalue effect is not conclusive: it sometimes helps, but sometimes doesn't. It is unclear how these explanations will guide student-teacher training. Thus, I think I will keep my review score.

---

> > > ### Author Response · Authors · 2023-08-11
> > > **Thanks for acknowledging our response!**
> > >
> > > Dear reviewer,
> > >
> > > We are grateful for your prompt response to our rebuttal!
> > > **Thank you for acknowledging that we addressed your question regarding the "double-edgedness". We sincerely hope you also found our response to your other two concerns helpful!**
> > >
> > > We also understand that it is unclear to you as to how our findings can end up providing concrete guidance for knowledge distillation (KD).  While we believe that this work has the potential to give rise to practical guidance in the long-term, we acknowledge that evaluating this can certainly be subjective.
> > >
> > > Nevertheless, we jot down some of our thoughts on this for the curious readers:
> > >
> > > 1. Without our finding, future empirical research may dedicate significant efforts into forcing the student to _precisely_ fit the teacher. **Our work warns that being pedantic about a precise fit would be counterproductive, thus potentially shaping empirical research.**
> > >
> > > 2. Our theoretical model opens the door to analyzing the many variations of KD. E.g., it is not hard to extend it to "progressive KD" with intermediate teacher checkpoints (https://arxiv.org/abs/2110.08532). An exciting possibility is that **one could (easily) brainstorm new KD objectives due to the simplicity of our theoretical framework**, with the hope of a *provably* better exaggerated bias. One could then empirically explore those variants in the deep learning setting. In this sense, our theory has the potential to inspire new practice.
> > >
> > > 3. As you note in the strengths, this is a first step towards bridging theoretical intuition and a remarkable practical phenomenon about KD. Such theory-practice work is always a challenging endeavor: deep learning practice is almost always messy, and any tractable theory almost always requires great simplifications. **We have gone great lengths to bridge this theory-practice disconnect:** we report our observations on a wide range of image/language settings, and we verify our theory in multiple settings where assumptions break.
> > >
> > > We hope this provides a glimpse of what we believe are _long-term_ value in our findings. But to reiterate, we understand that the value/confidence one may assign to these possibilities is subjective.
> > >
> > > Thanks again for engaging with our work and for your constructive feedback!
> > >
> > > Regards,
> > > Authors

---

### Official Review · Reviewer_vVNQ · 2023-07-09

**Soundness:** 3 good
**Presentation:** 3 good
**Contribution:** 3 good
**Rating:** 6
**Confidence:** 4

**Summary:**

This work delves into the understanding of knowledge distillation by studying the deviations between teacher and student models during the distillation process. It reveals two primary observations: students often underfit points that teachers find challenging, and the initial training phase is not crucial for distillation benefits as similar results can be achieved by switching from hard labels to teacher's soft predictions mid-training. To explain these observations, the authors propose two theoretical viewpoints: distillation acting as a regularizer in the eigenspace and as a denoiser of gradients. Empirical evidence supporting these theories was provided through experiments on various settings. Overall, the paper enriches our understanding of knowledge distillation, bridging the gap between theory and practice, while offering insights that could enhance the efficiency and effectiveness of knowledge distillation processes.

**Strengths:**

1. This paper conducts experiments across multiple model architectures and both vision and language tasks.
2. The observation is clearly presented and further explained from interesting perspectives.
3. This paper thoroughly reviews relevant literature and successfully positions itself within the context of existing research.

**Weaknesses:**

1. While Section 5.2 has presented the effect of teacher's interpolation level in CIFAR-100,  a synthetic dataset might solidify such findings further. It would be intriguing to explore whether varying the teacher's interpolation would correspondingly impact the degree of the student's exaggeration.

2. This paper focuses on the setting of distilling from logits, it would augment the paper's breadth if some analysis concerning feature distillation  were incorporated.

3. Not a weakness but the authors might find connection in this paper: Zhang et al., "Do Not Blindly Imitate the Teacher: Loss Perturbation for Knowledge Distillation". I understand the authors may not yet have had the opportunity to consider its relevance to this study given its recent release, but it appears to echo this paper's claim that not matching the teacher probabilities exactly can be a good thing.

**Questions:**

See above.

**Limitations:**

See above.

---

> ### Author Rebuttal · Authors · 2023-08-04
>
>
> Thank you for appreciating the breadth of our empirical findings and giving our paper a positive rating!
>
> ------
> > It would be intriguing to explore whether varying the teacher's interpolation would correspondingly impact the degree of the student's exaggeration.
>
> Interesting question! Based on our theory, the student’s exaggeration would arise independent of the teacher’s interpolation level. It is only the student’s generalization that is confounded by the teacher’s interpolation level.
>
> **Indeed, as evinced by Figure A2 in our global response PDF, under both the interpolated and non-interpolated CIFAR-100 teacher, the confidence exaggeration appears.**
>
> We’d also like to refer you to the plots of ImageNet in Fig 7 (where there’s no interpolation) where we do find confidence exaggeration.
>
> We're afraid we are not sure why a synthetic dataset would be necessary: even in CIFAR100, we are already able to manipulate the teacher's interpolation level by training it to different extents. Did you have something else in mind? Please let us know! Thanks!
>
> ----
>
> > 2. it would augment the paper's breadth if some analysis concerning feature distillation were incorporated.
>
> This is an exciting follow-up exploration. There may be many interesting effects that may arise when the exaggerated bias is induced at a feature level rather than at the logit level.
>
> But to gently push back on characterizing this as a weakness: given the existing poor understanding of distillation, we followed the spirit of work such as [1] in rigorously understanding at least the most widely used form of distillation. We believe this in itself is a significant first step. Furthermore, please note that, unlike prior works, we simultaneously attack this from both a theoretical and empirical viewpoint (across a breadth of settings) bridging the gap between the two.
>
>
> -----
>
> 3. Not a weakness but the authors might find connection in this paper: Zhang et al., "Do Not Blindly Imitate the Teacher: Loss Perturbation for Knowledge Distillation".  I understand the authors may not yet have had the opportunity to consider its relevance to this study given its recent release,...
>
> Thank you for bringing up this (contemporaneous) work. Interestingly, their rationale for not fitting the teacher is complementary to ours. Their rationale is that overfitting to the teacher may be bad since the teacher can be inaccurate. In our viewpoint, even if the teacher is 100% accurate, it helps to not overfit to the teacher. Our theory says that this will help ignore certain lower eigendirections picked up by the teacher.
>
> -------
>
> [1] Stanton et al., Does Knowledge Distillation Really Work?
>
> ------
>
> We hope our responses provide satisfactory answers to your questions. If so, we hope you will re-evaluate the paper in a more positive light. Thank you for your time!

---

### Official Review · Reviewer_MZ5g · 2023-07-11

**Soundness:** 3 good
**Presentation:** 3 good
**Contribution:** 3 good
**Rating:** 6
**Confidence:** 3

**Summary:**

The paper investigates one of the surprising findings in the field of knowledge distillation, which is that often times, the student deviates from the teacher in the process of mimicking it, and that some times this results in the student performing even better than the teacher (e.g., self-distillation). The authors claim that the reason this happens is that the distillation process will systematically try to exaggerate the confidence of the teacher: if the original teacher's confidence was low, the distilled student will have an even lower confidence, and if the original confidence was high, the student's confidence will be even higher. The paper then explains that there is another exaggeration which happens in the distillation process - exaggeration of bias - and how this can be understood as a cause for the former exaggeration (of confidence). The authors conduct experiments on many kinds of datasets (e.g., CIFAR, ImageNet) where they verify their findings. The takeaway from this work are explanations of some of the unintuitive behaviors of the distillation process, as well as some tips for practical purposes (to achieve better performance through the distilled student).

**Strengths:**

The paper is well written; the problem well motivated (why certain observations about knowledge distillation are surprising).

The authors have used the results of prior work properly to put forth a hypothesis which connects and expands on them (e.g., extending the result of Mobahi et al. to GD trained models). The mathematical proof connecting the two distinct properties - exaggeration in implicit bias in GD trained models and exaggeration in confidence is properly explained.

Results are shown which confirm their hypothesis on multiple different datasets.

**Weaknesses:**

Even though the individual sections of the paper are well written (see strengths), the paper as a whole does appear to be an amalgamation of many different sections, and it is a bit difficult to figure out what the overarching theme is. A better narrative would have been if things were progressively going from one point to the other - for example, exaggerating of the implicit bias is the root cause --> which then causes an exaggeration in confidence level --> which then explains why students perform better than the teacher sometimes. Right now, it is not clear where the narrative is leading to for the most part, until the conclusion section.

One of the important conclusions of the paper (line 259-260) - "distillation can hurt the student when the teacher does not achieve sufficient top-1 accuracy on the training data." - seems to be contrary to many of the other results that people have observed. For example, the paper that the authors themselves cite, Cho and Hariharan [1] mentions that the bigger (more accurate, as measured by top-1 accuracy) are often not the most appropriate models for performing knowledge distillation. The student's achieve better performance using smaller (less accurate teachers). How do the authors reconcile their results with [1]'s observations? Plus, I don't think this conclusion helps explain why self-distillation improves the performance of the new student, when the older student was less accurate (compared to the distilled student).




References

[1] On the Efficacy of Knowledge Distillation. Cho et al. ICCV 2019


**Post rebuttal update:**
I appreciate the rebuttal given by the authors and I'm increasing my rating by 1 point

**Questions:**

It has been shown that the efficacy of knowledge distillation depends a lot on certain practical considerations [2]. It will be good if the authors can expound their experiment setup (augmentations used, number of iterations etc.).


References

[2] Knowledge distillation: A good teacher is patient and consistent. Beyer et al. 2022



**Limitations:**

There is no section for limitations, and in fact I think it will be useful to have a section like this. It is particularly more useful in a field like knowledge distillation where often times some unintuitive behavior of the student is emerging. So, if there are certain observations that the presented work cannot explain (e.g., see the weaknesses section), please try to have a discussion about it.

---

> ### Author Rebuttal · Authors · 2023-08-04
>
> Thank you for your positive score on the paper, and for appreciating how we connect the two disjoint lines of research!
>
> --------
>
> > "distillation can hurt the student when the teacher does not achieve sufficient top-1 accuracy on the training data." - seems to be contrary to many of the other results that people have observed.
>
> This is a great question. **We explain why there is no contradiction here:**
>    1. First, a quick note: Our argument is also supported via orthogonal experiments in other papers [1, 2] who note that inaccurate teachers can hurt distillation. We have mentioned [1] already, but will also cite [2].
>    2.  When one increases teacher size as Cho and Hariharan do, there are two confounding variables which have opposing effects on distillation:
>        - Effect A: Teacher’s top-1 accuracy: this must help the student
>        - Effect B: Capacity mismatch, specifically the teacher’s non-target logits become too rich/complex: this must hurt the student
>
> **From Cho and Hariharan, we cannot conclude that Effect A (top-1 accuracy) hurts the student since even the opposing Effect B (capacity mismatch of non-target logits) is present**! In our work, we carefully isolate the effect of A by focusing on self-distillation. In self-distillation, intuitively, B becomes negligible since the student can represent the non-target logits of the teacher. In our controlled setting, we find that higher top-1 accuracy does help the student, thereby verifying the effect of A alone. Thus, we hope you appreciate the value in our controlled experiment which helps clarify an apparent contradiction in prior works.
>
> --------
>
> > it is a bit difficult to figure out what the overarching theme is…it is not clear where the narrative is leading to for the most part, until the conclusion section.
>
> We appreciate your valuable feedback about making the overarching story clearer upfront. We apologize for the lack of clarity on this. To clarify, currently our narrative follows the actual course of inquiry in our research (we began by understanding what deviations exist, and then how they may have emerged.). For now, we have two simpler ideas for fixing the issue you bring up:
> - Add a more explicit outline of our claim at the end of the introduction. In short: distillation exaggerates the implicit bias of GD $\implies$ This results in both (a) deviations from the teacher in the form of exaggerated confidence and (b) improved generalization (subject to other confounding factors). This reconciles how (a) can co-occur with (b).
> - We will also add a graphical model visualizing the above mechanism.
>
> Do you think this would help address your concern reasonably?
>
> --------
>
> > There is no section for limitations, and in fact I think it will be useful to have a section like this.
>
> **We would like to note that the first appendix section lays out the limitations in detail,** as Reviewer zo9s and tbs9 have noticed. But based on your feedback, we will make sure to refer to this in the main paper clearly.
>
> > It will be good if the authors can expound their experiment setup (augmentations used, number of iterations etc.).
>
> We would like to note that **Appendix C.1 and Table 1 cover all experiment details.** We made sure to use hyperparameters as recommended in prior work, while also providing key ablations in Appendix C.5 (for batch size, learning rate, training time, distillation weight, evaluation metric).
>
> --------
>
> **We hope that our key clarification that there is no contradiction in our findings with prior work, helps you re-evaluate the paper with a more positive score. Thanks again for your feedback!**
>
> [1]: Fotis Iliopoulos, Vasilis Kontonis, Cenk Baykal, Gaurav Menghani, Khoa Trinh, and Erik Vee. Weighted distillation with unlabeled examples.
>
> [2]: Zaida Zhou, Chaoran Zhuge, Xinwei Guan, and Wen Liu. Channel distillation: Channel-wise attention for knowledge distillation. CoRR, abs/2006.01683, 2020.

---

> > ### Comment · Reviewer_MZ5g · 2023-08-12
> > **Response to the rebuttal**
> >
> > I thank the authors for their rebuttal. I particularly appreciated their clarification on the apparent contradiction between their theory and Cho and Hariharan's work. I would encourage the authors to somehow include this in their paper, even if it is in the appendix. This is because this is one of the major "surprises" that people have noticed working with knowledge distillation, and this clarification that the authors have provided will help shed some light on it. I also thank the authors for pointing out the relevant details present elsewhere in their paper.
> > Overall, I am more confident with the submission and I am increasing my rating from 5 to 6.

---

> > > ### Author Response · Authors · 2023-08-12
> > > **Thank you!**
> > >
> > > Dear reviewer,
> > >
> > > Thank you for raising your score and acknowledging that our response addresses your concerns satisfactorily. We will certainly add an extended discussion regarding Cho and Hariharan to the paper. Thanks for raising a valuable question in your review!
> > >
> > > Authors

---

### Author Rebuttal · Authors · 2023-08-04

The response PDF contains Fig A2 as requested by Reviewer vVNQ (R2) and Fig A1 as requested by Reviewer zo9s (R4).

(Note: In case a link to the PDF is not visible, clicking on `Revisions` above should lead to a link.)

---

### Decision · Program_Chairs · 2023-09-21

**Decision:**

Accept (poster)

**Comment:**

This paper studies the learning behavior of knowledge distillation when the student model deviates from a teacher model. All the reviewers give accept recommendation. They appreciate the paper has clear motivation, studies a less-explored problem and provides solid experiment results and analysis. The reviewers also give suggestions for improving the paper quality, including the presentation and figures. The authors are encouraged to further improve the paper by following these suggestions. It is clear this paper meets the standard and thus AC recommends accept.